# Binary and analog variation of synapses between cortical pyramidal neurons

Sven Dorkenwald[1,2]*[†], Nicholas L Turner[1,2†], Thomas Macrina[1,2†], Kisuk Lee[1,3†], Ran Lu[1†], Jingpeng Wu[1†], Agnes L Bodor[4†], Adam A Bleckert[4†], Derrick Brittain[4†], Nico Kemnitz[1], William M Silversmith[1], Dodam Ih[1], Jonathan Zung[1], Aleksandar Zlateski[1], Ignacio Tartavull[1], Szi-Chieh Yu[1], Sergiy Popovych[1,2], William Wong[1], Manuel Castro[1], Chris S Jordan[1], Alyssa M Wilson[1], Emmanouil Froudarakis[5,6], JoAnn Buchanan[4], Marc M Takeno[4], Russel Torres[4], Gayathri Mahalingam[4], Forrest Collman[4], Casey M Schneider-Mizell[4], Daniel J Bumbarger[4], Yang Li[4], Lynne Becker[4], Shelby Suckow[4], Jacob Reimer[5,6], Andreas S Tolias[5,6,7], Nuno Macarico da Costa[4], R Clay Reid[4], H Sebastian Seung[1,2]*

[1]Princeton Neuroscience Institute, Princeton University, Princeton, United States; [2]Computer Science Department, Princeton University, Princeton, United States; [3]Brain & Cognitive Sciences Department, Massachusetts Institute of Technology, Cambridge, United States; [4]Allen Institute for Brain Science, Seattle, United States; [5]Department of Neuroscience, Baylor College of Medicine, Houston, United States; [6]Center for Neuroscience and Artificial Intelligence, Baylor College of Medicine, Houston, United States; [7]Department of Electrical and Computer Engineering, Rice University, Houston, United States

*For correspondence:
svenmd@princeton.edu (SD);
sseung@princeton.edu (HSS)

[†]These authors contributed equally to this work

**Abstract** Learning from experience depends at least in part on changes in neuronal connections. We present the largest map of connectivity to date between cortical neurons of a defined type (layer 2/3 [L2/3] pyramidal cells in mouse primary visual cortex), which was enabled by automated analysis of serial section electron microscopy images with improved handling of image defects (250 × 140 × 90 μm$^3$ volume). We used the map to identify constraints on the learning algorithms employed by the cortex. Previous cortical studies modeled a continuum of synapse sizes by a log-normal distribution. A continuum is consistent with most neural network models of learning, in which synaptic strength is a continuously graded analog variable. Here, we show that synapse size, when restricted to synapses between L2/3 pyramidal cells, is well modeled by the sum of a binary variable and an analog variable drawn from a log-normal distribution. Two synapses sharing the same presynaptic and postsynaptic cells are known to be correlated in size. We show that the binary variables of the two synapses are highly correlated, while the analog variables are not. Binary variation could be the outcome of a Hebbian or other synaptic plasticity rule depending on activity signals that are relatively uniform across neuronal arbors, while analog variation may be dominated by other influences such as spontaneous dynamical fluctuations. We discuss the implications for the longstanding hypothesis that activity-dependent plasticity switches synapses between bistable states.

## Editor's evaluation

Cortical synaptic plasticity mechanisms shape excitatory connectivity during learning and development. A long-standing question is whether these processes are determined by pre- and postsynaptic activity and whether the resulting synaptic changes result in a continuous, graded distribution of strengths. Dorkenwald and colleagues use extensive ultrastructural data to study cortical excitatory synaptic spines and demonstrate that the population is a very well-described discrete mix of "small"

and "large" connections, with graded variability around these dominant modes. Co-innervated connections result in strong correlations between the discrete small/large variable, but not the graded component, supporting a model in which correlated activity results in jumps between small and large synaptic strengths.

## Introduction

Synapses between excitatory neurons in the cortex and hippocampus are typically made onto spines, tiny thorn-like protrusions from dendrites (*Yuste, 2010*). In the 2000s, long-term in vivo microscopy studies showed that dendritic spines change in shape and size, and can appear and disappear (*Bhatt et al., 2009*; *Holtmaat and Svoboda, 2009*). Spine dynamics were interpreted as synaptic plasticity, because spine volume is well correlated with physiological strength of a synapse (*Matsuzaki et al., 2001*; *Noguchi et al., 2011*; *Holler et al., 2021*). The plasticity was thought to be in part activity-dependent, because spine volume increases with long-term potentiation (*Matsuzaki et al., 2004*; *Kopec et al., 2006*; *Noguchi et al., 2019*). Given that the sizes of other synaptic structures (postsynaptic density, presynaptic active zone, and so on) are well correlated with spine volume and with each other (*Harris and Stevens, 1989*), we use the catch-all term 'synapse size' to refer to the size of any synaptic structure, and 'synapse strength' as a synonym.

In the 2000s, some hypothesized the existence of 'learning spines' and 'memory spines', appearing to define two discrete categories that are structurally and functionally different (*Kasai et al., 2003*; *Bourne and Harris, 2007*). Quantitative studies of cortical synapses, however, found no evidence for discreteness (*Harris and Stevens, 1989*; *Arellano, 2007*; *Loewenstein et al., 2011*; *Loewenstein et al., 2015*; *de Vivo et al., 2017*; *Santuy et al., 2018*; *Kasai et al., 2021*). Whether in theoretical neuroscience or artificial intelligence, it is common for the synaptic strengths in a neural network model to be continuously variable, enabling learning to proceed by the accumulation of arbitrarily small synaptic changes over time.

Here, we reexamine the discrete versus continuous dichotomy using a wiring diagram between 334 layer 2/3 pyramidal cells (L2/3 PyCs) reconstructed from serial section electron microscopy (ssEM) images of mouse primary visual cortex. We show that synapses between L2/3 PyCs are well modeled as a binary mixture of log-normal distributions. If we further restrict consideration to dual connections, two synapses sharing the same presynaptic and postsynaptic cells, the binary mixture exhibits a statistically significant bimodality. It is therefore plausible that the binary mixture reflects two underlying structural states, and is more than merely an improvement in curve fitting.

According to our best fitting mixture model, synapse size is the sum of a binary variable and a log-normal continuous variable. To probe whether these variables are modified by synaptic plasticity, we examined dual connections. Previous analyses of dual connections examined pairs of synapses between the same axon and same dendrite branches (SASD) (*Sorra and Harris, 1993*; *Koester and Johnston, 2005*; *Bartol et al., 2015*; *Kasthuri et al., 2015*; *Dvorkin and Ziv, 2016*; *Bloss et al., 2018*; *Motta et al., 2019*). They found that such synapse pairs are correlated in size, and the correlations have been attributed to activity-dependent plasticity. In contrast, our population of synapse pairs includes distant synapses made on different branches and is constrained to one cell type (L2/3 PyC). We find that the binary variables are highly correlated, while the continuous variables are not. If we expand the analysis to include a broader population of cortical synapses, bimodality is no longer observed.

The specificity of our synaptic population was made possible because each of the 334 neurons taking part in the 1735 connections in our cortical wiring diagram could be identified as an L2/3 PyC based on a soma and sufficient dendrite and axon contained in the ssEM volume. The closest precedents for wiring diagrams between cortical neurons of a defined type had 29 connections between 43 L2/3 PyCs in mouse visual cortex (*Lee et al., 2016*), 63 connections between 22 L2 excitatory neurons in mouse medial entorhinal cortex (*Schmidt et al., 2017*), and 32 connections between 89 L4 neurons in mouse somatosensory cortex (*Motta et al., 2019*).

Our cortical reconstruction has been made publicly available and used concurrently in other studies (https://www.microns-explorer.org/phase1)(*Schneider-Mizell et al., 2021*; *Turner et al., 2022*). The code that generated the reconstruction is already freely available.

## Results

### Handling of ssEM image defects

We acquired a $250 \times 140 \times 90$ μm³ ssEM dataset (*Figure 1—figure supplement 1*) from L2/3 primary visual cortex of a P36 male mouse at $3.58 \times 3.58 \times 40$ nm³ resolution. When we aligned a pilot subvolume and applied state-of-the-art convolutional nets, we found many reconstruction errors, mainly due to misaligned images and damaged or incompletely imaged sections. This was disappointing given reports that convolutional nets can approach human-level performance on one benchmark ssEM image dataset (*Beier et al., 2017*; *Zeng et al., 2017*). The high error rate could be explained by the fact that image defects are difficult to escape in large volumes, though they may be rare in small (<1000 μm³) benchmark datasets.

Indeed, ssEM images were historically considered problematic for automated analysis (*Briggman and Bock, 2012*; *Lee et al., 2019*) because they were difficult to align, contained defects caused by lost or damaged serial sections, and had inferior axial resolution (*Knott et al., 2008*). These difficulties were the motivation for developing block face electron microscopy (bfEM) as an alternative to ssEM (*Denk and Horstmann, 2004*). Most large-scale ssEM reconstructions have been completely manual, while many large-scale bfEM reconstructions have been semi-automated (19/20 and 5/10 in Table 1 of *Kornfeld and Denk, 2018*). On the other hand, the higher imaging throughput of ssEM (*Nickell and Zeidler, 2019*; *Yin et al., 2019*) makes it suitable for scaling up to volumes that are large enough to encompass the arbors of mammalian neurons.

We supplemented existing algorithms for aligning ssEM images (*Saalfeld et al., 2012*) with human-in-the-loop capabilities. After manual intervention by a human expert, large misalignments were resolved but small ones still remained near damaged locations and near the borders of the volume. Therefore, we augmented the training data for our convolutional net with simulated misalignments and missing sections (*Figure 1a*, *Figure 1—figure supplement 2*). The resulting net was better able to trace neurites through such image defects (*Figure 1b*, quantification in *Figure 1—figure supplement 3*). Other methods for handling ssEM image defects are being proposed (*Li, 2019*), and we can look forward to further gains in automated reconstruction accuracy in the future.

### Wiring diagram between cells in L2/3

After alignment and automatic segmentation (Materials and methods), we semi-automatically identified 417 PyCs and 34 inhibitory cells with somas in the volume based on morphological characteristics and automated nucleus detection (*Figure 1d and e*, Materials and methods). We then chose a subset of 362 PyCs and 34 inhibitory cells with sufficient neurite length within the volume for proofreading. Remaining errors in the segmentation of these cells were corrected using an interactive system that enabled human experts to split and merge objects.

We estimate that the PyC reconstructions were corrected through ~1300 hr of human proofreading to yield 670 mm cable length (axon: 100 mm, dendrite: 520 mm, perisomatic: 40 mm, *Figure 1—figure supplement 2*). We examined 12 randomly sampled axons and conservatively estimated that 0.28 merge errors per millimeter remain after proofreading (see Materials and methods for other estimates). The dendrites of the PyCs receive more than one-quarter of the 3.2 million synapses that were automatically detected in the volume (Materials and methods, *Turner et al., 2020*). However, the synapses onto PyC dendrites are almost all from 'orphan' axons, defined as those axonal fragments that belong to somas of unknown location outside the volume. Using these automatically detected synapses as a starting point, we mapped all connections between this set of PyCs and inhibitory cells (Materials and methods). The end result was a wiring diagram of 6210 synapses from 3347 connections in the dataset. The subgraph of PyCs contained 1960 synapses from 1735 connections between 334 L2/3 PyCs (*Figure 2a*). Note that some connections are multisynaptic, that is, they are mediated by multiple synapses sharing the same presynaptic and postsynaptic cells (*Figure 2b*, *Figure 2—figure supplement 1*, see *Table 1* for a tabular overview of these statistics).

For clarity, we emphasize that our usage of the term 'multisynaptic' refers to multiple synapses between a single cell pair. A connection between two PyCs usually (89.1%) contains one synapse, but can contain up to five synapses (2: 9.22%, 3: 1.38%, 4: 0.17%, 5: 0.12%, *Figure 2c*) (these numbers should be taken with the caveat that the observed number of synapses for a connection is a lower bound for the true number of synapses, because two PyCs with cell bodies in our EM volume could synapse with each other outside the bounds of the volume.). In comparison, only 60.3% of connections

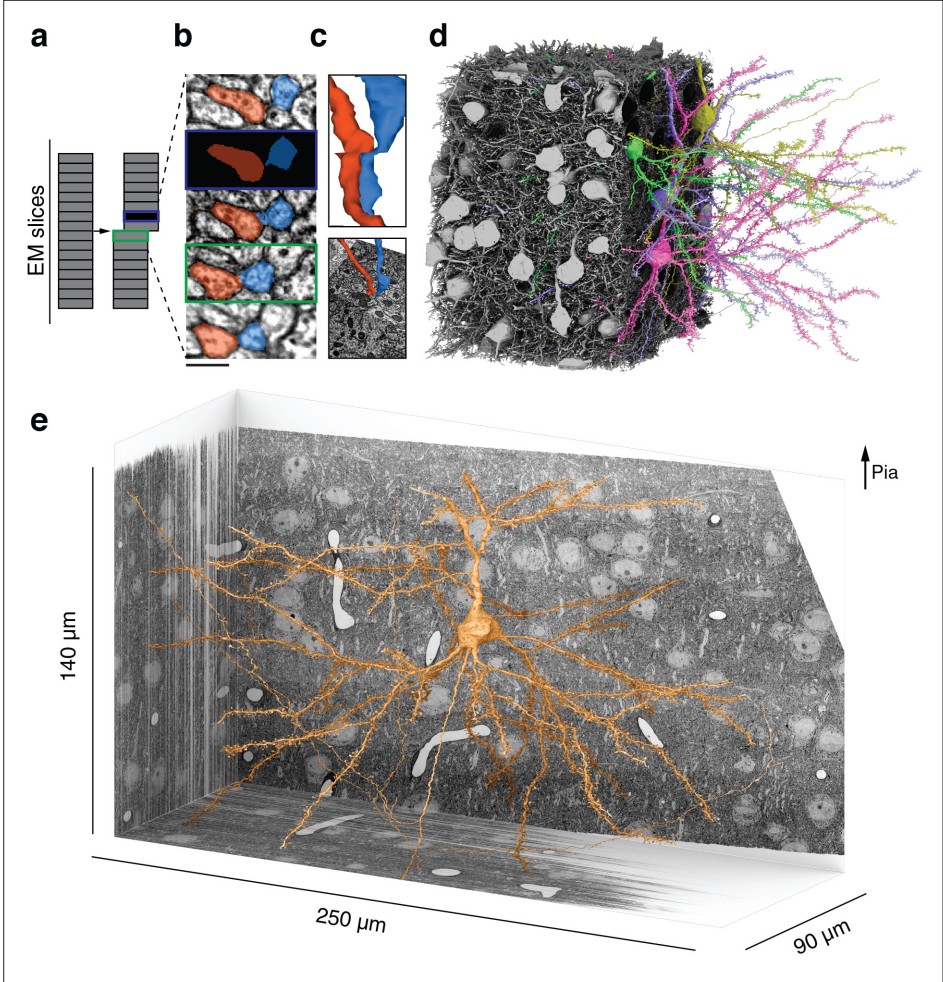

**Figure 1.** Reconstructing cortical circuits in spite of serial section electron microscopy (ssEM) image defects. (**a**) Ideally, imaging serial sections followed by computational alignment would create an image stack that reflects the original state of the tissue (left). In practice, image stacks end up with missing sections (blue) and misalignments (green). Both kinds of defects are easily simulated when training a convolutional net to detect neuronal boundaries. Small subvolumes are depicted rather than the entire stack, and image defects are typically local rather than extending over an entire section. (**b**) The resulting net can trace more accurately, even in images not previously seen during training. Here, a series of five sections contains a missing section (blue frame) and a misalignment (green). The net 'imagines' the neurites through the missing section, and traces correctly in spite of the misalignment. (**c**) 3D reconstructions of the neurites exhibit discontinuities at the misalignment, but are correctly traced. (**d**) All 362 pyramidal cells with somas in the volume (gray), cut away to reveal a few examples (colors). (**e**) Layer 2/3 (L2/3) pyramidal cell reconstructed from ssEM images of mouse visual cortex. Scale bars: 300 nm (**b**).

The online version of this article includes the following figure supplement(s) for figure 1:

**Figure supplement 1.** Reconstruction of connections between layer 2/3 (L2/3) pyramidal cells.

**Figure supplement 2.** Examples of reconstructed neurites near image defects.

**Figure supplement 3.** Quantitative evidence for the effectiveness of training data augmentation.

---

from PyCs on inhibitory cells were monosynaptic. Similarly, 62.1% connections made by inhibitory neurons were monosynaptic when targeting other inhibitory neurons, which reduces to only 42.6% when targeting PyCs. While the number of synapses per PyC-PyC connection varies least compared to the other three categories, we observed the highest variance in synapse sizes for these connections (*Figure 2d and e*). Here, we quantified synaptic cleft size as the number of voxels labeled by the output of our automated cleft detector (*Figure 2—figure supplement 2*). The dimensions of our reconstructions allowed us to observe dual connections with two synapses more than 100 μm

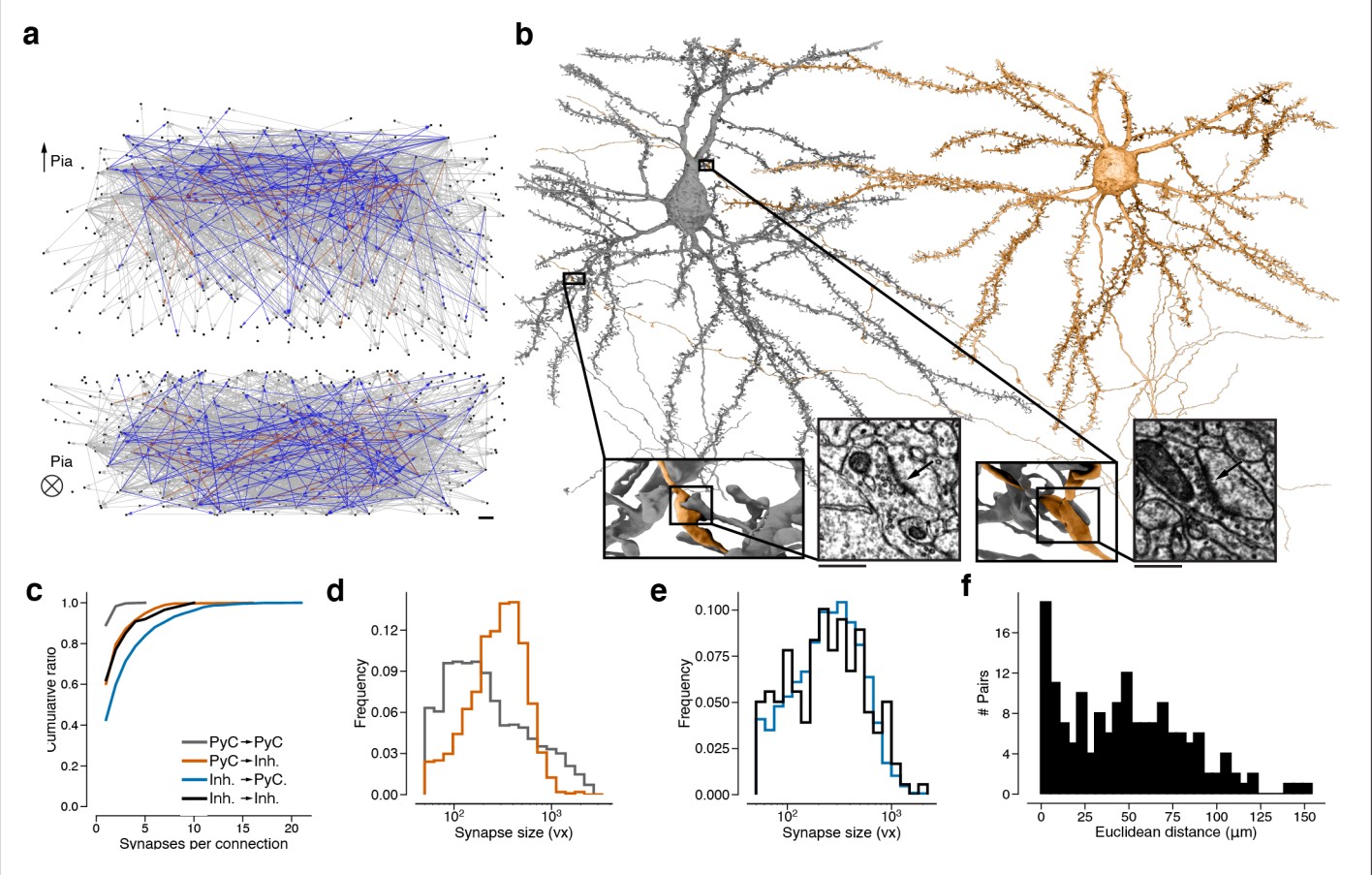

**Figure 2.** Wiring diagram for cortical neurons including multisynaptic connections. (**a**) Wiring diagram of 362 proofread layer 2/3 (L2/3) pyramidal cells (PyCs) as a directed graph. Two orthogonal views with nodes at 3D locations of cell bodies. Single (gray), dual (blue), and triple, quadruple, quintuple (red) connections. (**b**) Dual connection from a presynaptic cell (orange) to a postsynaptic cell (gray). Ultrastructure of both synapses can be seen in closeups from the electron microscopy (EM) images. The Euclidean distance between the synapses is 64.3 μm. (**c**) Normalized distributions of synapses sizes for L2/3 PyCs synapses separated by postsynaptic cell type. (**d**) Same as (**c**) for inhibitory cells in layer 2/3. (**e**) Cumulative distributions of the number of synapses per connection for different pre- and postsynaptic cell types. (**f**) Distribution of Euclidean distances between synapse pairs of dual connections. Median distance is 46.5 μm. Scale bars: 10 μm (**a**), 500 nm (**b**).

The online version of this article includes the following figure supplement(s) for figure 2:

**Figure supplement 1.** Renderings of all synapses from multisynaptic connections between layer 2/3 (L2/3) pyramidal cells.

**Figure supplement 2.** Examples of synapses between layer 2/3 (L2/3) pyramidal cells.

apart (**Figure 2b and f**), involving different axonal and dendritic branches. Previous analyses reporting correlations between synapses from dual synaptic connections only included synapses that were close to another and were between the SASD.

## Binary latent states

Previous studies of cortical synapses have found a continuum of synapse sizes (**Arellano, 2007**) that is approximated by a log-normal distribution (**Loewenstein et al., 2011**; **de Vivo et al., 2017**; **Santuy et al., 2018**; **Kasai et al., 2021**). Even researchers who report bimodally distributed synapse size on a log-scale in hippocampus (**Spano et al., 2019**) still find log-normally distributed synapse size in neocortex (**de Vivo et al., 2017**) by the same methods.

We quantified the size of each synapse by the volume of the spine head (**Figures 2b and 3a**) (spine head volume excludes the spine neck, which is at most only weakly correlated in size with other synaptic structures [**Arellano, 2007**]). In the following, 'spine volume' will serve as a synonym for spine head volume. Spine volumes spanned over two orders of magnitude, though 75% of spines lie within a single order of magnitude. The distribution of spine volumes is highly skewed, with a long tail of

**Table 1.** Overview of number of data points obtained in this study.

| | |
|---|---|
| Number of L2/3 PyCs in dataset | 417 |
| Number of L2/3 PyCs selected for proofreading | 362 |
| Number of proofread L2/3 PyCs connecting to any other L2/3 PyCs | 334 |
| Number of inhibitory cells in dataset | 34 |
| Number of synapses (automated) in the dataset | 3,239,275 |
| Number of outgoing synapses (automated) in the dataset from proofread L2/3 PyCs | 10,788 |
| Number of synapses between L2/3 PyCs | 1960 |
| Number of connections between L2/3 PyCs | 1735 |
| Number of connections between L2/3 PyCs with one synapse | 1546 |
| Number of connections between L2/3 PyCs with two synapses | 160 |
| Number of connections between L2/3 PyCs with three synapses | 24 |
| Number of connections between L2/3 PyCs with four synapses | 3 |
| Number of connections between L2/3 PyCs with five synapses | 2 |

large spines (*Figure 3b*) as observed before (*Loewenstein et al., 2011*; *Santuy et al., 2018*; *Kasai et al., 2021*). Because of the skew, it is helpful to visualize the distribution using a logarithmic scale for spine volume (*Loewenstein et al., 2011*; *Bartol et al., 2015*). We were surprised to find that the distribution deviated from normality, due to a 'knee' on the right side of the histogram (*Figure 3c*) (multiple researchers have proposed dynamical models of spine size that are consistent with approximately log-normal stationary distributions [*Kasai et al., 2021*]). A mixture of two normal distributions was a better fit than a single normal distribution when accounting for the number of free parameters (likelihood ratio test: p<1e-39, $n$=1960, Materials and methods).

We next restricted our consideration to the 320 synapses belonging to 160 dual connections between the PyCs. Again, a binary mixture of normal distributions was a better fit (*Figure 3d*, see *Figure 3—figure supplement 1* for linear plots) than a single normal distribution (normal fit not shown, likelihood ratio test: p<1e-7, $n$=320). Next, we made use of the fact that synapses from dual connections are paired. For each pair, we computed the geometric mean (i.e., mean in log-space) of spine volumes and found that this quantity is also well modeled by a binary mixture of normal distributions (*Figure 3e*, see *Figure 3—figure supplement 2* for the arithmetic mean, *Figure 3—figure supplement 3* for histograms without model fits and *Table 2* for fit results).

A binary mixture model might merely be a convenient way of approximating deviations from normality. We would like to know whether the components of our binary mixture could correspond to two structural states of synapses. A mixture of two normal distributions can be unimodal or bimodal, depending on the model parameters (for example, if the two normal distributions have the same weight and standard deviation, then the mixture is unimodal if and only if the separation between the means is at most twice the standard deviation) (*Robertson and Fryer, 1969*). When comparing best fit unimodal and bimodal mixtures we found that a bimodal model yields a significantly superior fit for spine volume and geometric mean of spine volume (p=0.0425, $n$=320; *Figure 3—figure supplement 4*, see *Holzmann and Vollmer, 2008*, for statistical methods).

A binary mixture model can be interpreted in terms of a binary latent variable. According to such an interpretation, synapses are drawn from two latent states (*Figure 3f*). In 'S' and 'L' states, spine volumes are drawn from log-normal distributions with small and large means, respectively. It should be noted that there is some overlap between mixture components (*Figure 3f*), so that an S synapse can be larger than an L synapse.

To validate this finding with a different measurement of synapse size, the number of voxels labeled by the output of our automated cleft detector. We found a close relationship between spine volume and cleft size in our data (*Figure 3—figure supplement 5a*), in accord with previous studies (*Harris and Stevens, 1989*; *Arellano, 2007*; *Bartol et al., 2015*). When spine volume is replaced by cleft size in the preceding analysis, we obtain similar results (*Figure 3—figure supplement 5*).

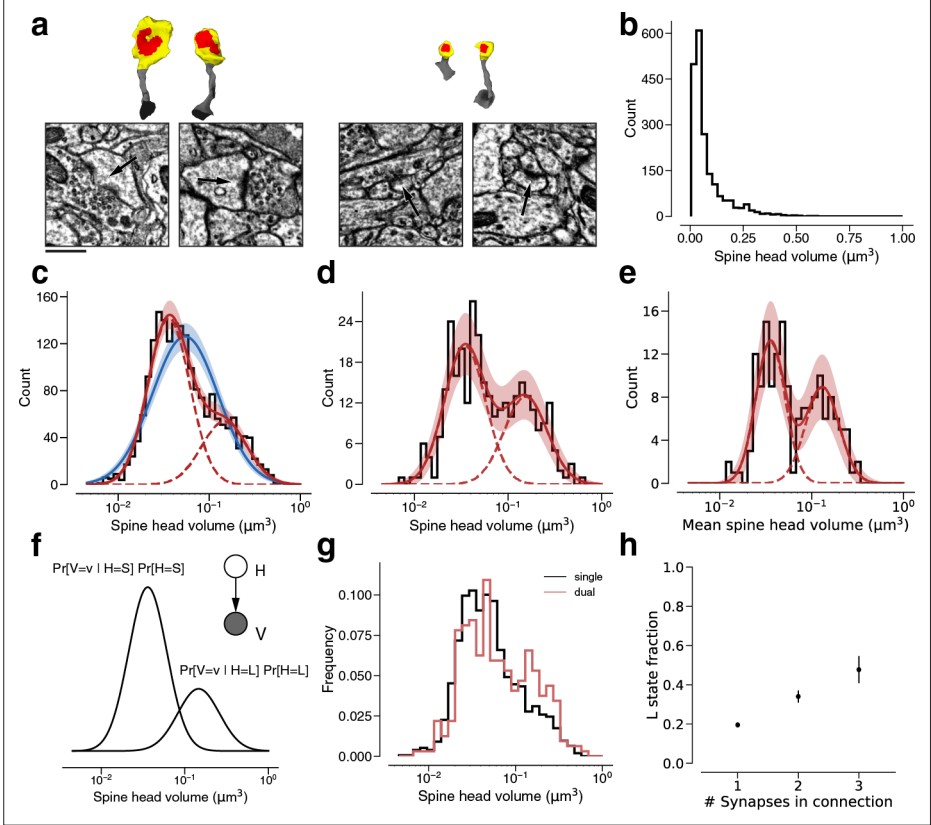

**Figure 3.** Modeling spine head volume with a mixture of two log-normal distributions. (**a**) Dendritic spine heads (yellow) and clefts (red) of dual connections between layer 2/3 pyramidal cells (L2/3) PyCs. The associated electron microscopy (EM) cutout shows a 2D slice through the synapse. The synapses are centered in the EM images. (**b**) Skewed histogram of spine volume for all 1960 recurrent synapses between L2/3 PyCs, with a long tail of large spines. (**c**) Histogram of the spine volumes in (**b**), logarithmic scale. A mixture (red, solid) of two log-normal distributions (red, dashed) fits better (likelihood ratio test, p<1e-39, n=1960) than a single normal (blue). (**d**) Spine volumes belonging to dual connections between L2/3 PyCs, modeled by a mixture (red, solid) of two log-normal distributions (red, dashed). (**e**) Dual connections between L2/3 PyCs, each summarized by the geometric mean of two spine volumes, modeled by a mixture (red, solid) of two log-normal distributions (red, dashed). (**f**) Mixture of two normal distributions as a probabilistic latent variable model. Each synapse is described by a latent state *H* that takes on values 'S' and 'L' according to the toss of a biased coin. Spine volume *V* is drawn from a log-normal distribution with mean and variance determined by latent state. The curves shown here represent the best fit to the data in (**d**). Heights are scaled by the probability distribution of the biased coin, known as the mixture weights. (**g**) Comparison of spine volumes for single (black) and dual (red) connections. (**h**) Probability of the 'L' state (mixture weight) versus number of synapses in the connection. Error bars are standard deviations estimated by bootstrap sampling. Scale bar: 500nm (**a**). Error bars are $\pm\sqrt{n}$ of the model fit (**c, d, e**) and standard deviation from bootstrapping (**h**).

The online version of this article includes the following figure supplement(s) for figure 3:

**Figure supplement 1.** Linear spine head volume distributions.

**Figure supplement 2.** Arithmetic means.

**Figure supplement 3.** Fits versus raw data histograms.

**Figure supplement 4.** Modeling spine volume with a bimodal versus unimodal mixture of two normal distributions.

**Figure supplement 5.** Modeling cleft size with a mixture of two normal distributions.

**Figure supplement 6.** Synapse size by connection type.

**Figure supplement 7.** Relation of dendritic spine volume to spine apparatus.

**Table 2.** Overview of results from log-normal mixture fits for different synapse subpopulations.

| Subset of L2/3 L2/3 PyC synapses | S | | | L | | | |
| --- | --- | --- | --- | --- | --- | --- | --- |
| | Mean (log$_{10}$ µm³) | Std (log$_{10}$ µm³) | Weight | Mean (log$_{10}$ µm³) | Std (log$_{10}$ µm³) | Weight | N |
| All synapses | −1.42 | 0.24 | 0.77 | −0.77 | 0.22 | 0.23 | 1960 |
| Single synapses | −1.41 | 0.24 | 0.81 | −0.76 | 0.21 | 0.19 | 1546 |
| Dual synapses | −1.44 | 0.23 | 0.64 | −0.77 | 0.21 | 0.36 | 320 |
| Triple synapses | −1.49 | 0.17 | 0.36 | −0.86 | 0.30 | 0.64 | 72 |
| All synapses with weights refitted to single synapses | (−1.42) | (0.24) | 0.80 | (−0.77) | (0.248) | 0.20 | 1960 and 1546 |
| All synapses with weights refitted to dual synapses | (−1.42) | (0.24) | 0.66 | (−0.77) | (0.248) | 0.34 | 1960 and 320 |
| All synapses with weights refitted to triple synapses | (−1.42) | (0.24) | 0.52 | (−0.77) | (0.248) | 0.48 | 1960 and 72 |
| Geometric mean of dual synapses | −1.44 | 0.16 | 0.58 | −0.87 | 0.18 | 0.42 | 160 |
| Arithmetic mean of dual synapses | −1.43 | 0.16 | 0.53 | −0.85 | 0.18 | 0.47 | 160 |

According to our two-state model, the parameters of the mixture components should stay roughly constant for the distribution of any subset of synapses between L2/3 PyCs. To probe model dependence on the number of synapses per connection, we individually fit a Gaussian mixture to the population of synapses from single, dual, and triple connections and found that their mixture components were not significantly different. Parameter estimates for these fits lie within sampling error of the single connection dataset (*Figure 3—figure supplement 6*). When comparing these distributions we observed an overrepresentation of large synapses for dual connections compared to single connections (*Figure 3g*). We wondered if the previously reported mean spine volume increase with the number of synapses per connection (*Figure 3—figure supplement 6*, *Bloss et al., 2018*) could be explained with a synapse redistribution between the latent states. This time, we only fit the component weights to single, dual, triple connections while keeping the Gaussian components constant (see Materials and methods). We found a linear increase in fraction of synapses in the 'L' state with the number of synapses per connection (*Figure 3h*). (This relationship was found for the *observed* number of synapses. On average, this number is expected to increase with the *true* number of synapses. Therefore, mean spine volume is also expected to increase with the true number of synapses per connection)

Large spines have been reported to contain an intracellular organelle called a spine apparatus (SA), which is a specialized form of smooth endoplasmic reticulum (ER) (*Peters and Kaiserman-Abramof, 1970*; *Spacek, 1985*; *Harris and Stevens, 1989*). We manually annotated SA in all dendritic spines of all synapses between L2/3 PyCs, and confirmed quantitatively that the probability of an SA increases with spine volume (*Figure 3—figure supplement 7*, Materials and methods).

## Correlations at dual connections

Positive correlation between synapse sizes at dual connections has been reported previously in hippocampus (*Sorra and Harris, 1993*; *Bartol et al., 2015*; *Bloss et al., 2018*) and neocortex (*Kasthuri et al., 2015*; *Motta et al., 2019*) for synapse pairs formed by the same axonal and dendritic branches. According to our binary mixture model, synapse size is the sum of a binary variable and a log-normal continuous variable. We decided to quantify the contributions of these variables to synapse size correlations.

The dendritic spines for all dual connections between L2/3 PyCs are rendered in *Figure 2—figure supplement 1*. A positive correlation between the two spine volumes of each dual connection is evident in a scatter plot of the spine volume pairs (*Figure 4a*, see *Figure 4—figure supplement 1* for an unoccluded plot; Pearson's *r*=0.418). We fit the joint distribution of the spine volumes by a mixture model like *Figure 3f*, while allowing the latent states to be correlated (*Figure 4a and f*, see *Table 3* for fit results, *Figure 4—figure supplement 2* for the same analysis for synaptic cleft sizes). In the

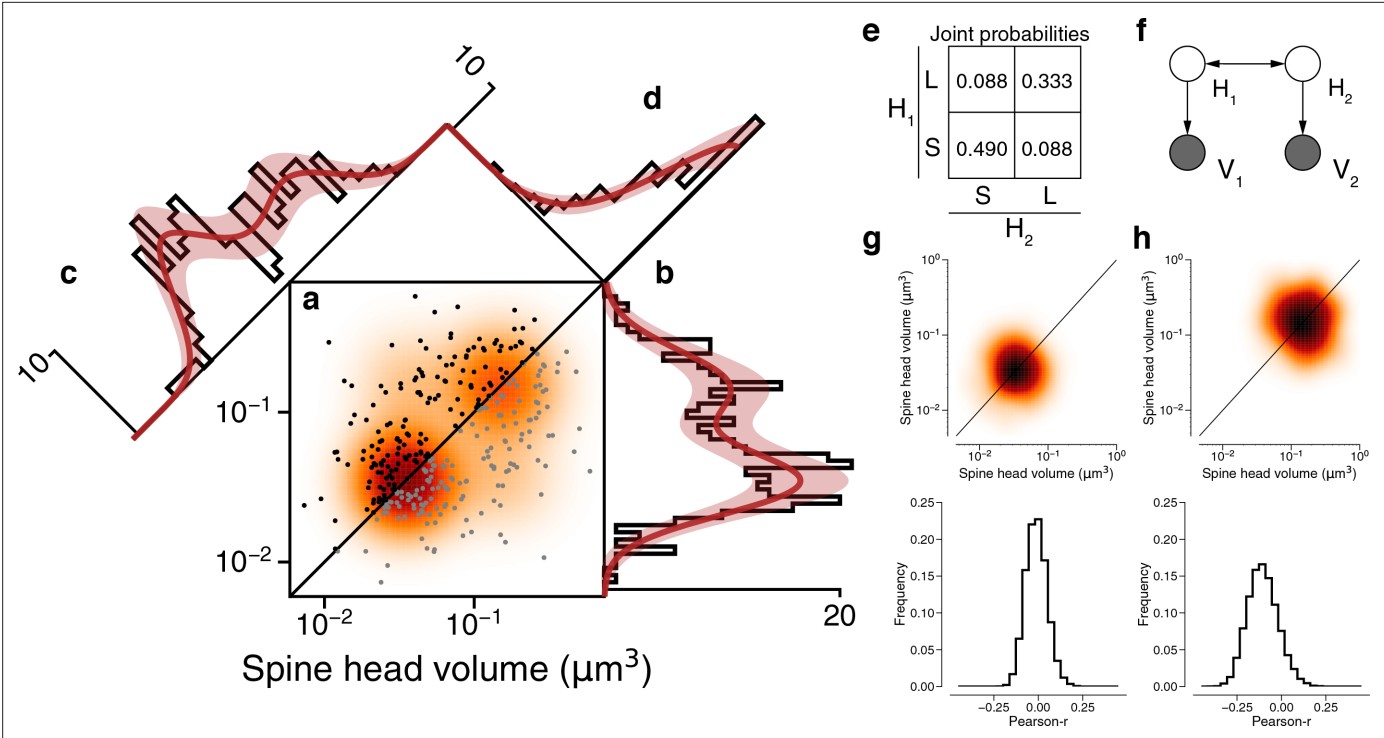

**Figure 4.** Latent state correlations between spines at dual connections. (**a**) Scatter plot of spine volumes (black, lexicographic ordering) for dual connections. Data points are mirrored across the diagonal (gray). The joint distribution is fit by a mixture model (orange) like that of **Figure 3f**, but with latent states correlated as in (**e**). (**b**) Projecting the points onto the vertical axis yields a histogram of spine volumes for dual connections (**Figure 3d**). Model is derived from the joint distribution. (**c**) Projecting onto the $x=y$ diagonal yields a histogram of the geometric mean of spine volumes (**Figure 3e**). Model is derived from the joint distribution. (**d**) Projecting onto the $x=-y$ diagonal yields a histogram of the ratio of spine volumes. (**e**) The latent states of synapses in a dual connection ($H_1$ and $H_2$) are more likely to be the same (SS or LL) than different (SL/LS), as shown by the joint probability distribution. (**f**) When conditioned on the latent states, the spine volumes ($V_1$ and $V_2$) are statistically independent, as shown in this dependency diagram of the model. (**g**), (**h**) Sampling synapse pairs to SS and LL states according to their state probabilities. The top shows a kernel density estimation of multiple iterations of sampling. The bottom shows the distribution of Pearson's $r$ correlations across many sampling rounds ($N=10,000$). Error bars are $\pm\sqrt{n}$ of the model fit.

The online version of this article includes the following figure supplement(s) for figure 4:

**Figure supplement 1.** Fits versus raw distributions.

**Figure supplement 2.** Latent state correlations between clefts at dual connections.

**Figure supplement 3.** Residuals spine head volume after subtracting binary components.

**Figure supplement 4.** Synapses in a dual connection: near versus far pairs.

**Figure supplement 5.** Dual connection correlations are not a result of axon or dendrite biases.

**Figure supplement 6.** Removing constraints on the synaptic population eliminates bimodality and reduces correlations.

**Table 3.** Overview of results from hidden Markov model (HMM) log-normal component fits for different dual synaptic connection subpopulations.

| Subset of L2/3 L2/3 PyC dual synaptic connections | S | | L | | Weights | | | | |
| --- | --- | --- | --- | --- | --- | --- | --- | --- | --- |
| | Mean (log$_{10}$ µm$^3$) | Std (log$_{10}$ µm$^3$) | Mean (log$_{10}$ µm$^3$) | Std (log$_{10}$ µm$^3$) | SS | SL+ LS | LL | Pearson's phi | N |
| All connections | −1.470 | 0.216 | −0.833 | 0.244 | 0.490 | 0.177 | 0.333 | 0.637 | 160 |
| Dist <median dist | −1.506 | 0.212 | −0.861 | 0.243 | 0.427 | 0.232 | 0.342 | 0.534 | 80 |
| Dist >median dist | −1.449 | 0.207 | −0.818 | 0.251 | 0.529 | 0.123 | 0.348 | 0.745 | 80 |

best-fitting model, SS occurs roughly half the time, LL one-third of the time, and the mixed states (SL, LS) occur more rarely (*Figure 4e*). The low probability of the mixed states can be seen directly in the scarcity of points in the upper left and lower right corners of the scatter plot (*Figure 4a*). Pearson's phi coefficient, the specialization of Pearson's correlation coefficient to binary variables, is 0.637.

Our mixture model assumes that the spine volumes are independent when conditioned on the latent states. To visualize whether this assumption is justified by the data, *Figure 4* shows 1D projections of the joint distribution onto different axes. The projection onto the vertical axis (*Figure 4b*) is the marginal distribution, the overall size distribution for all synapses that belong to dual connections (same as *Figure 3d*). The projection onto the $x=y$ diagonal (*Figure 4c*) is the distribution of the geometric mean of spine volume for each dual connection (same as *Figure 3e*). The projection onto the $x=-y$ diagonal (*Figure 4d*) is the distribution of the ratio of spine volumes for each dual connection. For all three projections, the good fit suggests that the data are consistent with the mixture model's assumption of isotropic normal distributions for the LL and SS states. (The $x=y$ and vertical histograms look bimodal because they are different projections of the same two 'bumps' in the joint distribution. If the probability of the mixed state (LS/SL) were high, there would be two additional off-diagonal bumps in the joint distribution, and the $x=y$ diagonal histogram would acquire another peak in the middle. In reality the probability of the mixed state is low, so the $x=y$ diagonal histogram is well modeled by two mixture components. The widths of the bumps are the same in both projections, but the distance between the bumps is longer in the $x=y$ diagonal histogram by a factor of root two. This explains why the mixture components are better separated in the distribution of geometric means (*Figure 3e*) than in the marginal distribution (*Figure 3d*), and hence why the statistical significance of bimodality is stronger for the geometric means.)

For a quantitative test of the isotropy assumption, we resampled observed spine volume pairs with weightings computed from the posterior probabilities of the SS and LL states (*Figure 4g and h*). If the model were consistent with the data, the resampled data would obey an isotropic normal distribution. Indeed, Pearson's correlation for the resampled data is not significantly greater than zero (*Figure 4g and h*). Therefore, the spine volumes in a dual connection are approximately uncorrelated when conditioned on the latent states. We validated this result by examining the residual synapse sizes after subtracting the binary components and found no remaining correlation between then synapse pairs (*Figure 4—figure supplement 3*).

## Specificity of latent state correlations

Could the observed correlations between synapses in dual connections be caused by crosstalk between plasticity of neighboring synapses (<10 µm separation), which has been reported previously (*Harvey and Svoboda, 2007*; *Harvey et al., 2008*)? We looked for dependence of latent state correlations on separation by splitting dual connections into two groups, those with synapses nearer or farther than the median Euclidean distance in the volume of 46.5 µm. Both groups were fit by mixture models with positive correlations between latent variables (near: $\varphi = 0.53$, far: $\varphi = 0.75$, see Materials and methods, *Figure 4—figure supplement 4*). In other words, for dual connections involving pairs of distant synapses, the latent state correlations are still strong.

We also considered the possibility of correlations in pairs of synapses sharing the same presynaptic cell but not the same postsynaptic cell, or pairs of synapses sharing the same postsynaptic cell but not the same presynaptic cell (*Bartol et al., 2015*; *Kasthuri et al., 2015*; *Dvorkin and Ziv, 2016*; *Bloss et al., 2018*; *Motta et al., 2019*). We randomly drew such synapse pairs from the set of synapses that belong to dual connections (and hence belong to PyCs that participate in dual connections). Correlations in the latent state or synapse size were negligible (same axon: $\varphi = -0.11\pm0.08$ SD, $r = -0.06\pm0.06$ SD; same dendrite: $\varphi = -0.06\pm0.06$ SD, $r = -0.13\pm0.05$ SD; *Figure 4—figure supplement 5*), similar to previous findings (*Bloss et al., 2018*; *Motta et al., 2019*).

## Discussion

Our synapse size correlations are specific to pairs of synapses that share both the same presynaptic and postsynaptic L2/3 PyCs, similar to previous findings (*Sorra and Harris, 1993*; *Koester and Johnston, 2005*; *Bartol et al., 2015*; *Kasthuri et al., 2015*; *Dvorkin and Ziv, 2016*; *Bloss et al., 2018*; *Motta et al., 2019*). We have further demonstrated that the correlations exist even for large spatial

separations between synapses. More importantly, we have shown the correlations are confined to the binary latent variables in our synapse size model; the log-normal analog variables exhibit little or no correlation.

The correlations in the binary variables could arise from a Hebbian or other synaptic plasticity rule driven by presynaptic and postsynaptic activity signals that are relatively uniform across neuronal arbors. Such signals are shared by synapses in a multisynaptic connection (*Sorra and Harris, 1993*; *Koester and Johnston, 2005*; *Bartol et al., 2015*; *Kasthuri et al., 2015*; *Dvorkin and Ziv, 2016*; *Bloss et al., 2018*; *Motta et al., 2019*).

We speculate that much of the analog variation arises from the spontaneous dynamical fluctuations that have been observed at single dendritic spines through time-lapse imaging. Computational models of this temporal variance suggest that it can account for much of the population variance (*Yasumatsu et al., 2008*; *Loewenstein et al., 2011*; *Statman et al., 2014*). Experiments have shown that large dynamical fluctuations persist even after activity is pharmacologically blocked (*Yasumatsu et al., 2008*; *Statman et al., 2014*; *Sando et al., 2017*; *Sigler, 2017*). Another possibility is that the analog variation arises from plasticity driven by activity-related signals that are local to neighborhoods within neuronal arbors.

It remains unclear whether the binary latent variable in our model reflects some underlying bistable mechanism or is merely a convenient statistical description. Our latent variable model is consistent with the scenario in which synapses behave like binary switches that are flipped by activity-dependent plasticity. Switch-like behavior could arise from bistable networks of molecular interactions at synapses (*Lisman, 1985*), has been observed in physiology experiments on synaptic plasticity (*Petersen et al., 1998*; *O'Connor et al., 2005*), and has been the basis of a number of computational models of memory (*Tsodyks, 1990*; *Amit and Fusi, 1994*; *Fusi et al., 2005*). In this scenario, synapses only appear volatile due to fluctuations in the analog variable (*Loewenstein et al., 2011*), which obscures an underlying bistability.

In a second scenario, the bimodality of synapse size does not reflect an underlying bistability. For example, models of activity-dependent plasticity can cause synapses to partition into two clusters located at upper and lower bounds for synaptic size (*Song et al., 2000*; *van Rossum et al., 2000*; *Rubin et al., 2001*). In this scenario, synapses are intrinsically volatile, and bimodality arises because learning drives them to extremes.

We would like to suggest that the first scenario of binary switches is somewhat more plausible, for two reasons. First, it is unclear how the second scenario could lead to strong correlations in the binary variable. Second, it is unclear how the second scenario could be consistent with the little or no correlation that remains in our data once the contribution from the binary latent variables is removed. This argument is tentative; more experimental and theoretical studies are needed to draw firmer conclusions.

Bimodality and strong correlations were found for a restricted ensemble of synapses, those belonging to dual connections between L2/3 PyCs. However, bimodality is not observed for the ensemble of all excitatory synapses onto L2/3 PyCs, including those from orphan axons (*Figure 4—figure supplement 6*). This ensemble is similar to ones studied previously, that is, synapses onto L2/3 PyCs (*Arellano, 2007*), L4 neurons (*Motta et al., 2019*), or L5 PyCs (*Loewenstein et al., 2011*). Bimodality and strong correlations are also not observed for the ensemble of all dual connections received by L2/3 PyCs, including those from orphan axons (*Figure 4—figure supplement 6*). Because our findings are based on a highly specific population of synapses, they are not inconsistent with previous studies that failed to find evidence for discreteness of cortical synapses (*Harris and Stevens, 1989*; *Arellano, 2007*; *Loewenstein et al., 2011*; *Loewenstein et al., 2015*; *de Vivo et al., 2017*; *Santuy et al., 2018*).

Why does the bimodality disappear when one includes dual connections with orphan axons? In our view, the simplest explanation is that this is due to the fact that orphan axons come from a mixed population of cell types, each one with its different distribution of synapse sizes. While each cell type to cell type connection might have its unique properties, they are lost to the observer when combining connections between different cell types together.

Bimodality and correlations may turn out to be heterogeneous across classes of neocortical synapses. Heterogeneity in the hippocampus has been demonstrated by the finding that dual connections onto granule cell dendrites in the middle molecular layer of dentate gyrus (*Bromer et al., 2018*)

do not exhibit the correlations that are found in stratum radiatum of CA1 (*Bartol et al., 2015*; *Bloss et al., 2018*).

Since the physiological strength of a multisynaptic connection can be approximately predicted from the sum of synaptic sizes (*Holler-Rickauer et al., 2019*), our S and L latent states and their correlations have implications for the debate over whether infrequent strong connections play a disproportionate role in cortical computation (*Song et al., 2005*; *Cossell et al., 2015*; *Scholl, 2019*).

## Materials and methods

### Mouse

All procedures were in accordance with the Institutional Animal Care and Use Committees at the Baylor College of Medicine and the Allen Institute for Brain Science. Same sex littermates were housed together in individual cages with one to four mice per cage. Mice were maintained on a regular diurnal lighting cycle (12:12 light:dark) with ad libitum access to food and water and nesting material for environmental enrichment. Mice were housed in the Taub Mouse Facility of Baylor College of Medicine, accredited by AAALAC (The Association for Assessment and Accreditation of Laboratory Animal Care International). The animal used for this experiment was healthy and not involved in any previous procedure or experiment.

### Mouse line

Functional imaging was performed in a transgenic mouse expressing fluorescent GCaMP6f. For this dataset, the mouse we used was a triple heterozygote for the following three genes: (1) Cre driver: CamKIIa-Cre (Jax: 005359 https://www.jax.org/strain/005359), (2) tTA driver: B6;CBA-Tg(Camk2a-tTA)1Mmay/J (Jax: 003010 https://www.jax.org/strain/003010), (3) GCaMP6f Reporter: Ai93 (Allen Institute).

### Cranial window surgery

Anesthesia was induced with 3% isoflurane and maintained with 1.5–2% isoflurane during the surgical procedure. Mice were injected with 5–10 mg/kg ketoprofen subcutaneously at the start of the surgery. Anesthetized mice were placed in a stereotaxic head holder (Kopf Instruments) and their body temperature was maintained at 37°C throughout the surgery using a homeothermic blanket system (Harvard Instruments). After shaving the scalp, bupivicane (0.05 cc, 0.5%, Marcaine) was applied subcutaneously, and after 10–20 min an approximately 1 cm$^2$ area of skin was removed above the skull and the underlying fascia was scraped and removed. The wound margins were sealed with a thin layer of surgical glue (VetBond, 3 M), and a 13 mm stainless-steel washer clamped in the headbar was attached with dental cement (Dentsply Grip Cement). At this point, the mouse was removed from the stereotax and the skull was held stationary on a small platform by means of the newly attached headbar. Using a surgical drill and HP 1/2 burr, a 3 mm craniotomy was made centered on the primary visual cortex (V1; 2.7 mm lateral of the midline, contacting the lambda suture), and the exposed cortex was washed with ACSF (125 mM NaCl, 5 mM KCl, 10 mM glucose, 10 mM HEPES, 2 mM CaCl$_2$, 2 mM MgSO$_4$). The cortical window was then sealed with a 3 mm coverslip (Warner Instruments), using cyanoacrylate glue (VetBond). The mouse was allowed to recover for 1–2 hr prior to the imaging session. After imaging, the washer was released from the headbar and the mouse was returned to the home cage.

### Widefield imaging

Prior to two-photon imaging, we acquired a low-magnification image of the 3 mm craniotomy under standard illumination.

### Two-photon imaging

Imaging for candidate mice was performed in V1, in a 400 × 400 × 200 µm$^3$ volume with the superficial surface of the volume at the border of L1 and L2/3, approximately 100 µm below the pia. Laser excitation was at 920 nm at 25–45 mW depending on depth. The objective used was a 25× Nikon objective with a numerical aperture of 1.1, and the imaging point spread function was measured

with 500 nm beads and was approximately 0.5 × 0.5 × 3 $\mu m^3$ in *x*, *y*, and *z*. Pixel dimensions of each imaging frame were 256×256.

## Tissue preparation and staining

The protocol of *Hua et al., 2015*, was combined with the protocol of *Tapia et al., 2012*, to accommodate a smaller tissue size and to improve TEM contrast. Mice were transcardially perfused with 2.5% paraformaldehyde and 1.25% glutaraldehyde. After dissection, 200 µm thick coronal slices were cut with a vibratome and post-fixed for 12–48 hr. Following several washes in CB (0.1 M cacodylate buffer pH 7.4), the slices were fixed with 2% osmium tetroxide in CB for 90 min, immersed in 2.5% potassium ferricyanide in CB for 90 min, washed with deionized (DI) water for 2× 30 min, and treated with freshly made and filtered 1% aqueous thiocarbohydrazide at 40°C for 10 min. The slices were washed 2× 30 min with DI water and treated again with 2% osmium tetroxide in water for 30 min. Double washes in DI water for 30 min each were followed by immersion in 1% aqueous uranyl acetate overnight at 4°C. The next morning, the slices in the same solution were placed in a heat block to raise the temperature to 50°C for 2 hr. The slices were washed twice in DI water for 30 min each, and then incubated in Walton's lead aspartate pH 5.0 for 2 hr at 50°C in the heat block. After another double wash in DI water for 30 min each, the slices were dehydrated in an ascending ethanol series (50%, 70%, 90%, 100%×3) 10 min each and two transition fluid steps of 100% acetonitrile for 20 min each. Infiltration with acetonitrile:resin dilutions (2p:1p, 1p:1p and 2p:1p) were performed on a gyratory shaker overnight for 4 days. Slices were placed in 100% resin for 24 hr followed by embedding in Hard Plus resin (EMS, Hatfield, PA). Slices were cured in a 60°C oven for 96 hr. The best slice based on tissue quality and overlap with the 2p region was selected.

## Sectioning and collection

A Leica EM UC7 ultramicrotome and a Diatome 35-degree diamond ultra-knife were used for sectioning at a speed of 0.3 mm/s. Eight to ten serial sections were cut at 40 nm thickness to form a ribbon, after which the microtome thickness setting was set to 0 in order to release the ribbon from the knife. Using an eyelash probe, pairs of ribbons were collected onto copper grids covered by 50 nm thick LUXEL film.

## Transmission electron microscopy

We made several custom modifications to a JEOL-1200EXII 120 kV transmission electron microscope (*Yin et al., 2019*). A column extension and scintillator magnified the nominal field of view by 10-fold with negligible loss of resolution. A high-resolution, large-format camera allowed fields of view as large as (13 µm)$^2$ at 3.58 nm resolution. Magnification reduced the electron density at the phosphor, so a high-sensitivity sCMOS camera was selected and the scintillator composition tuned to generate high-quality EM images with exposure times of 90–200 ms. Sections were acquired as a grid of 3840 × 3840 px images ('tiles') with 15% overlap.

## Alignment in two blocks

The dataset was divided by sections into two blocks (1216 and 970 sections), with the first block containing substantially more folds. Initial alignment and reconstruction tests proceeded on the second block of the dataset. After achieving satisfactory results, the first block was added, and the whole dataset was further aligned to produce the final 3D image. The alignment process included stitching (assembling all tiles into a single image per section), rough alignment (aligning the set of section images with one affine per section), coarse alignment (nonlinear alignment on lower resolution data), and fine alignment (nonlinear alignment on higher resolution data).

## Alignment, block one

The tiles of the first block were stitched into one montaged image per section and rough aligned using a set of customized and automated modules based on the 'TrakEM2' (*Cardona et al., 2012*) and 'Render' (*Zheng et al., 2018*) software packages.

### Stitching

After acquisition, a multiplicative intensity correction based on average pixel intensity was applied to the images followed by a lens distortion of individual tiles using nonlinear transformations (*Kaynig*

*et al., 2010*). Once these corrections were applied, correspondences between tiles within a section were computed using SIFT features, and each tile was modeled with a rigid transform.

### Rough alignment

Using 20× downsampled stitched images, neighboring sections were roughly aligned (*Saalfeld et al., 2012*). Correspondences were again computed using SIFT features, and each section was modeled with a regularized affine transform (90% affine+10% rigid), and all correspondences and constraints were used to generate the final model of one affine transform per tile. These models were used to render the final stitched section image into rough alignment with block two.

## Alignment, block two

The second block was stitched and aligned using the methods of *Saalfeld et al., 2012*, as implemented in Alembic (*Macrina and Ih, 2019*).

### Stitching

For each section, tiles containing tissue without clear image defects were contrast normalized by centering the intensities at the same location in each tile, stretching the overall distribution between the 5th and 95th intensity percentiles. During imaging, a 20× downsampled overview image of the section was also acquired. Each tile was first placed according to stage coordinates, approximately translated based on normalized cross-correlation (NCC) with the overview image, and then finely translated based on NCC with neighboring tiles. Block matching was performed in the regions of overlap between tiles using NCC with 140 px block radius, 400 px search radius, and a spacing of 200 px. Matches were manually inspected with 1× coverage, setting per-tile-pair thresholds for peak of match correlogram, distance between first and second peaks of match correlograms, and correlogram covariance, and less frequently, targeted match removal. A graphical user interface was developed to allow the operator to fine-tune parameters on a section-by-section basis, so that a skilled operator completed inspection in 40 hr. Each tile was modeled as a spring mesh, with nodes located at the center of each blockmatch operation, spring constants 1/100th of the constant for the between-tile springs, and the energy of all spring meshes within a section were minimized to a fractional tolerance of $10^{-8}$ using nonlinear conjugate gradient. The final render used a piecewise affine model defined by the mesh before and after relaxation, and maximum intensity blending.

### Rough alignment

Using 20× downsampled images, block matching between neighboring sections proceeded using NCC with 50 px block radius, 125 px search radius, and 250 px spacing. Matches were computed between nearest neighbor section pairs, then filtered manually in 8 hr. Correspondences were used to develop a regularized affine model per section (90% affine+10% rigid), which was rendered at full image resolution.

### Coarse alignment

Using 4× downsampled images, NCC-based block matching proceeded 300 px block radius, 200 px search radius, and 500 px spacing. Matches were computed between nearest and next-nearest section pairs, then manually filtered by a skilled operator in 24 hr. Each section was modeled as a spring mesh with spring constants 1/10th of the constant for the between-section springs, and the energy of all spring meshes within the block were minimized to a fractional tolerance of $10^{-8}$ using nonlinear conjugate gradient. The final render used a piecewise affine model defined by the mesh.

### Fine alignment

Using 2× downsampled images, NCC-based block matching proceeded 200 px block radius, 113 px search radius, and 100 px spacing. Matches were computed between nearest and next-nearest section pairs, then manually filtered by a skilled operator in 24 hr. Modeling and rendering proceeded as with coarse alignment, using spring constants were 1/20th of the constant for the between-section springs.

### Alignment, whole dataset

Blank sections were inserted manually between sections where the cutting thickness appeared larger than normal (11). The alignment of the whole dataset was further refined using the methods of *Saalfeld et al., 2012*, as implemented in Alembic (*Macrina and Ih, 2019*).

## Coarse alignment

Using 64× downsampled images, NCC-based block matching proceeded 128 px block radius, 512 px search radius, and 128 px spacing. Matches were computed between neighboring and next-nearest neighboring sections, as well as 24 manually identified section pairs with greater separation, then manually inspected in 70 hr. Section spring meshes had spring constants 1/20th of the constant for the between-section springs. Mesh relaxation was completed in blocks of 15 sections, 5 of which were overlapping with the previous block (2 sections fixed), each block relaxing to a fractional tolerance of $10^{-8}$. Rendering proceeded similarly as before.

## Fine alignment

Using 4× downsampled images, NCC-based block matching proceeded 128 px block radius, 512 px search radius, and 128 px spacing. Matches were computed between the same section pairs as in coarse alignment. Matches were excluded only by heuristics. Modeling and rendering proceeded similar to coarse alignment, with spring constants 1/100th the constant for the between-section springs. Rendered image intensities were linearly rescaled in each section based on the 5th and 95th percentile pixel values.

### Image volume estimation

The imaged tissue has a trapezoidal shape in the sectioning plane. Landmark points were placed in the aligned images to measure this shape. We report cuboid dimensions for simplicity and comparison using the trapezoid midsegment length. The original trapezoid has a short base length of 216.9 μm, long base length of 286.2 μm, and height 138.3 μm. The imaged data has 2176 sections, which measures 87.04 μm with a 40 nm slice thickness.

### Image defect handling

Cracks, folds, and contaminants were manually annotated as binary masks on 256× downsampled images, dilated by 2 px, then inverted to form a defect mask. A tissue mask was created using nonzero pixels in the 256× downsampled image, then eroded by 2 px to exclude misalignments at the edge of the image. The image mask is the union of the tissue and defect masks, and it was upsampled and applied during the final render to set pixels not included in the mask to zero. We created a segmentation mask by excluding voxels that had been excluded by the image mask for three consecutive sections. The segmentation mask was applied after affinity prediction to set affinities not included in the mask to zero.

### Affinity prediction

Human experts used VAST (*Berger et al., 2018*) to manually segment multiple subvolumes from the current dataset and a similar dataset from mouse V1. Annotated voxels totaled 1.29 billion at full image resolution.

We trained a 3D convolutional network to generate 3 nearest neighbor (*Turaga et al., 2010*) and 13 long-range affinity maps (*Lee, 2017*). Each long-range affinity map was constructed by comparing an equivalence relation (*Jain et al., 2010*) of pairs of voxels spanned by an 'offset' edge (to preceding voxels at distances of 4, 8, 12, and 16 in *x* and *y*, and 2, 3, 4 in *z*). Only the nearest neighbor affinities were used beyond inference time; long-range affinities were used solely for training. The network architecture was modified from the 'Residual Symmetric U-Net' of *Lee, 2017*. We trained on input patches of size 128 × 128 × 20 at 7.16 × 7.16 × 40 nm³ resolution. The prediction during training was bilinearly upsampled to full image resolution before calculating the loss.

Training utilized synchronous gradient updates computed by four Nvidia Titan X Pascal GPUs each with a different input patch. We used the AMSGrad variant (*Reddi et al., 2019*) of the Adam optimizer (*Kingma and Ba, 2014*), with PyTorch's default settings except step size parameter $\alpha$=0.001.

We used the binary cross-entropy loss with 'inverse margin' of 0.1 *Huang and Jain, 2013*; patch-wise class rebalancing (*Lee, 2017*) to compensate for the lower frequency of boundary voxels; training data augmentation including flip/rotate by 90 degrees, brightness and contrast perturbations, warping distortions, misalignment/missing section simulation, and out-of-focus simulation (*Lee, 2017*); and lastly several new types of data augmentation including the simulation of lost section and co-occurrence of misalignment/missing/lost section.

Distributed computation of affinity maps used chunkflow (*Wu et al., 2019*). The computation was done with images at $7.16 \times 7.16 \times 40$ nm$^3$ resolution. The whole volume was divided into $1280 \times 1280 \times 140$ chunks overlapping by $128 \times 128 \times 10$, and each chunk was processed as a task. The tasks were injected into a queue (Amazon Web Service Simple Queue Service). For 2.5 days, 1000 workers (Google Cloud n1-highmem-4 with 4 vCPUs and 26 GB RAM, deployed in Docker image using Kubernetes) fetched and executed tasks from the queue as follows. The worker read the corresponding chunk from Google Cloud Storage using CloudVolume (*Silversmith et al., 2021*), and applied previously computed masks to black out regions with image defects. The chunk was divided into $256 \times 256 \times 20$ patches with 50% overlap. Each patch was processed to yield an affinity map using PZNet, a CPU inference framework (*Popovych, 2020*). The overlapping output patches were multiplied by a bump function, which weights the voxels according to the distance from patch center, for smooth blending and then summed. The result was cropped to $1024 \times 1024 \times 120$ vx and then previously computed segmentation masks were applied (see Image defect handling above).

## Watershed and size-dependent single linkage clustering

The affinity map was divided into $514 \times 514 \times 130$ chunks that overlapped by 2 voxels in each direction. For each chunk we ran a watershed and clustering algorithm (*Zlateski and Seung, 2015*) with special handling of chunk boundaries. If the descending flow of watershed terminated prematurely at a chunk boundary, the voxels around the boundary were saved to disk so that domain construction could be completed later on. Decisions about merging boundary domains were delayed, and information was written to disk so decisions could be made later. After the chunks were individually processed, they were stitched together in a hierarchical fashion. Each level of the hierarchy processed the previously delayed domain construction and clustering decisions in chunk interiors. Upon reaching the top of the hierarchy, the chunk encompassed the entire volume, and all previously delayed decisions were completed.

## Mean affinity agglomeration

The watershed supervoxels and affinity map were divided into $513 \times 513 \times 129$ chunks that overlapped by 1 in each direction. Each chunk was processed using mean affinity agglomeration (*Lee, 2017*; *Funke et al., 2019*). Agglomeration decisions at chunk boundaries were delayed, and information about the decisions was saved to disk. After the chunks were individually processed, they were combined in a hierarchical fashion similar to the watershed process.

## Training with data augmentations

We performed preliminary experiments on the effect of training data augmentation by simulating image defects on the publicly available SNEMI3D challenge dataset (http://brainiac2.mit.edu/SNEMI3D). We partitioned the SNEMI3D training volume of $1024 \times 1024 \times 100$ voxels into the center crop of $512 \times 512 \times 100$ voxels for validation, and the rest for training. Then we trained three convolutional nets to detect neuronal boundaries, one without any data augmentation ('baseline'), and the other two with simulated missing section ('missing section') and simulated misalignment ('misalignment') data augmentation, respectively. After training the three nets, we measured the robustness of each net to varying degrees of simulated image defects on the validation set (*Figure 1—figure supplement 3*). In the first measurement, we simulated a misalignment at the middle of the validation volume with varying numbers of pixel displacement. In the second measurement, we introduced varying numbers of consecutive missing sections at the middle of the validation volume. For each configuration of simulation, we ran an inference pipeline with the three nets to produce respective segmentations, and computed the variation of information error metric to measure the quality of the segmentations. For the measurement against simulated misalignment, we applied connected components to recompute the ground truth segmentation after introducing a misalignment, such that we

separated a single object into two distinct objects if the object is completely broken by the misalignment (e.g. the displacement of misalignment larger than the diameter of neurite).

## Synaptic cleft detection

Synaptic clefts were annotated by human annotators within a 310.7 μm³ volume, which was split into 203.2 μm³ training, 53.7 μm³ validation, and 53.7 μm³ test sets. We trained a version of the Residual Symmetric U-Net (*Lee, 2017*) with 3 downsampling levels instead of 4, 90 feature maps at the third downsampling instead of 64, and 'resize' upsampling rather than strided transposed convolution. Images and labels were downsampled to 7.16 × 7.16 × 40 nm³ image resolution. To augment the training data, input patches were transformed by (1) introducing misalignments of up to 17 pixels, (2) blacking out up to five sections, (3) blurring up to five sections, (4) warping, (5) varying brightness and contrast, and (6) flipping and rotating by multiples of 90 degrees. Training used PyTorch (*Paszke et al., 2017*) and the Adam optimizer (*Kingma and Ba, 2014*). The learning rate started from $10^{-3}$, and was manually annealed three times (505k training updates), before adding 67.2 μm³ of extra training data for another 670k updates. The extra training data focused on false positive examples from the network's predictions at 505k training updates, mostly around blood vessels. The trained network achieved 93.0% precision and 90.9% recall in detecting clefts of the test set. This network was applied to the entire dataset using the same distributed inference setup as affinity map inference. Connected components of the thresholded network output that were at least 50 voxels at 7.16 × 7.16 × 40 nm³ resolution were retained as predicted synaptic clefts.

## Synaptic partner assignment

Presynaptic and postsynaptic partners were annotated for 387 clefts, which were split into 196, 100, and 91 examples for training, validation, and test sets. A network was trained to perform synaptic partner assignment via a voxel association task (*Turner et al., 2020*). Architecture and augmentations were the same as for the synaptic cleft detector. Test set accuracy was 98.9% after 710k training iterations. The volume was separated into non-overlapping chunks of size 7.33 × 7.33 × 42.7 μm³ (1024 × 1024 × 1068 voxels), and the net was applied to each cleft in each chunk. This yielded a single prediction for interior clefts. For a cleft that crossed at least one boundary, we chose the prediction from the chunk which contained the most voxels of that cleft. Cleft predictions were merged if they connected the same synaptic partners and their centers-of-mass were within 1 μm. This resulted in 3,556,643 final cleft predictions.

## PyC proofreading

The mean affinity graph of watershed supervoxels was stored in our PyChunkedGraph backend, which uses an octree to provide spatial embedding for fast updates of the connected component sets from local edits. We modified the Neuroglancer frontend (*Maitin-Shepard et al., 2019*) to interface with this backend so users directly edit the agglomerations by adding and removing edges in the supervoxel graph (merge and split agglomerations). Connected components of this graph are meshed in chunks of supervoxels, and chunks affected by edits are updated in real time so users can always see a 3D representation of the current segmentation. Using a keypoint for each object (e.g. soma centroid), objects are assigned the unique ID of the connected component for the supervoxel which contains that location. This provides a means to update the object's ID as edits are made.

Cell bodies in the EM volume were semi-automatically identified. PyCs were identified by morphological features, including density of dendritic spines, presence of apical and basal dendrites, direction of main axon trunk, and cell body shape. We selected a subset of the 417 PyCs for proofreading based on the amount of visible neurite within the volume. A team of annotators used the meshes to detect errors in dendritic trunks and axonal arbors, then to correct those errors with 50,000 manual edits in 1044 person-hours. After these edits, PyCs were skeletonized, and both the branch and end points of these skeletons were identified automatically (with false negative rates of 1.7% and 1.4%, as estimated by annotators). Human annotators reviewed each point to ensure no merge errors and extend split errors where possible (210 person-hours). Putative broken spines targeted by PyCs were identified by selecting objects that received one or two synapses. Annotators reviewed, and attached these with 174 edits in 24 person-hours. Some difficult mergers came from small axonal boutons merged to

dendrites. We identified these cases by inspecting any predicted presynaptic site that resided within 7.5 µm of a postsynaptic site of the same cell, and corrected them with 50 person-hours.

## Estimation of final error rates

After proofreading was complete, a single annotator inspected 12 PyCs and spent 18 hr to identify all remaining errors in dendritic trunks and axonal arbors. The PyC proofreading protocol was designed to correct all merge errors, though not necessarily correct split errors caused by a masked segmentation. So this error estimation includes all merge errors identified and only split errors caused by less than three consecutive sections of masked segmentation. For 18.7 mm of dendritic path length inspected, three false splits (falsely excluding 160 synapses) and three false merges (falsely including 117 synapses) were identified (99% precision and 99% recall for incoming synapses). For 3.6 mm of axonal path length inspected, two false splits (falsely excluding four synapses) and one false merge (falsely including nine synapses) were identified (98% precision and 99% recall for outgoing synapses). We also sampled four dendritic branches with a collective 0.7 mm of path length, and identified 126 false negative and 0 false positive spines (88% recall of spines).

## PyC-PyC synapse proofreading

Synapses between PyC were extracted from the automatically detected and assigned synapses. We reviewed these synapses manually with 2× redundancy (1972 correct synapses out of 2433 putative synapses). Two predicted synapses out of these were 'merged' with other synaptic clefts. These cases were excluded from further analysis. One synapse was incorrectly assigned to a PyC and removed from the analysis. One other synapse was 'split' into two predictions, and these predictions were merged for analysis. We were not able to calculate spine head volumes for 8 out of these 1968 synapses and they were excluded from the analysis. This left 1960 synapses admitted into the analysis.

## Synapses from other excitatory axons

We randomly sampled synapses onto the PyCs and evaluated whether they are excitatory or inhibitory based on their shape, appearance, and targeted compartment ($n$=881 single excitatory synapses). We randomly sampled connections of two synapses onto PyCs and evaluated whether their presynaptic axon is excitatory or inhibitory and checked for reconstruction errors. Here, we manually checked that the automatically reconstructed path between the two synapses along the 3D mesh of the axon was error free ($n$=446 pairs of excitatory synapses). Those axons were allowed to contain errors elsewhere and we did not proofread any axons to obtain these pairs.

## Dendritic spine heads

We extracted a $7.33 \times 7.33 \times 4 \ \mu m^3$ cutout around the centroid of each synapse. The postsynaptic segment within that cutout was skeletonized using kimimaro (https://github.com/seung-lab/kimimaro; *Silversmith and Wu, 2022*), yielding a set of paths traveling from a root node to each leaf. The root node was defined as the node furthest from the synapse coordinate. Skeleton nodes participating in fewer than three paths were labeled as 'spine' while others were labeled as 'shaft'. The shaft labels were dilated along the skeleton until either (1) the distance to the segment boundary of the next node was more than 50 nm less than that of the closest (shaft) branch point, or (2) dilation went 200 nodes beyond the branch point. Each synapse was associated with its closest skeleton node, and a contiguous set of 'spine' labeled nodes. We finally separated spine head from neck by analyzing the distance to the segment boundary (DB) moving from the root of the spine to the tip. After segmenting the spine from the rest of the segment, we chose two anchor points: (1) the point with minimum DB value across the half of the spine toward the dendritic shaft and (2) the point with maximum DB value across the other half. A cut point was defined as the first skeleton node moving from anchor 1 to anchor 2 whose DB value was greater than $\frac{1}{3} DB_{anchor1} + \frac{2}{3} DB_{anchor2}$. Accounting for slight fluctuations in the DB value, we started the scan for the cut point at the closest node to anchor 2 that had DB value less than $\frac{1}{5} DB_{anchor1} + \frac{4}{5} DB_{anchor2}$. The skeleton of the spine head was defined as the nodes beyond this cut point to a leaf node, and the spine head mesh was defined as all spine mesh vertices which were closest to the spine head skeleton. The mesh of each head was identified as the subset of the postsynaptic segment mesh whose closest skeleton node was contained within the nodes labeled as

spine head. We then estimated the volume of this spine head by computationally sealing this mesh and computing its volume.

We identified poor extractions by computing the distance between each synapse centroid and the nearest node of its inferred spine head mesh. We inspected each inferred spine head for which this distance was greater than 35 nm, and corrected the mesh estimates of mistakes by relabeling mesh vertices using a 3D voronoi tessellation of points placed by a human annotator.

### Endoplasmic reticulum

We manually evaluated all spine heads between PyCs admitted to the analysis for whether they contained an SA, ER that is not an SA (ER), or none (no ER). We required the presence of at least two (usually parallel) membrane saccules for SA. Dense plate/region (synaptopodin and actin) in-between membrane saccules was an indicator. We found SA in spine heads and spine necks. We considered single lumens of organelles connecting to the ER network in the shaft as ER. We required that every ER could be traced back to the ER network in the dendritic shaft.

### Mixture models

Spine volumes and synapse sizes were $\log_{10}$-transformed before statistical modeling. Maximum likelihood estimation for a binary mixture of normal distributions used the expectation-maximization algorithm as implemented by Pomegranate (*Schreiber, 2017*). The algorithm was initialized using the k-means algorithm with the number of clusters set to equal the number of mixture components. For cleft size, the normal distributions were truncated at a lower bound of $\log_{10}(50)$ voxels, the same cutoff used in cleft detection. The truncation was implemented by modifications to the source code of Pomegranate. In this mixture model, each fitted distribution is parameterized with a mean, standard deviation, and weight per mixture component. In the case of two components we also refer to the weights as S and L state fractions. We used the square root of the estimated counts as errors on the fitted distributions (*Figure 3c, d and e*). To estimate errors on the state weights, we bootstrapped the population of synapses and reported the standard deviation of the fitted weights.

### Hidden Markov models

The joint distribution at dual connections was fitted by hidden Markov models (HMMs) with two latent states and emission probabilities given by normal distributions as described in the previous section. In total this resulted in four state probabilities (SS, SL, LS, LL). HMMs are trained on ordered pairs. Because there is no inherent order of the synapse pair from dual connections, we included each synapse pair twice in the dataset, one for each order.

### Correlation analysis

We assigned state probabilities to each dual synaptic pair using the best fit HMM. The following was done for SS and LL states independently. In each sampling iteration (*n*=10,000) we assigned individual synapse pairs to the state in question based on independent biased coin flips weighted by their respective state probability. For every such obtained sample we computed the Pearson's correlation of the sampled population of synapses (*Figure 4g and h*). For visualization in *Figure 4g and h* we applied a kernel density estimation (bw = 0.15 in $\log_{10}$-space).

### Parametric test for bimodality

For binary mixtures of normal distributions, the parameter regimes for bimodal and unimodal behaviors are known (*Robertson and Fryer, 1969*). The likelihood ratio of the best-fitting bimodal and unimodal models can be used for model selection (*Holzmann and Vollmer, 2008*). Mixture models were fit using Sequential Least Squares Programming using constraints on the parameter regimes for unimodal fits. We computed p-values using Chernoff's extension to boundary points of hypothesis sets (*Chernoff, 1954*) of Wilks' theorem governing asymptotics of the likelihood ratio (*Wilks, 1938*).

### Skeletonization

We developed a skeletonization algorithm similar to *Sato et al., 2000*, that operates on meshes. For each connected component of the mesh graph, we identify a root and find the shortest path to the farthest node. This procedure is repeated after invalidating all mesh nodes within the proximity of

the visited nodes until no nodes are left to visit. We make our implementation available through our package MeshParty (https://github.com/sdorkenw/MeshParty; *Dorkenwald et al., 2020*).

## Estimation of path lengths

We skeletonized all PyCs and labeled their first branch points close to the soma according to the compartment type of the downstream branches (axon, dendrite, ambiguous). If no branch point existed in close proximity a point at similar distance was placed. All skeleton nodes downstream from these nodes seen from the soma were labeled according to these labels. This allowed us to estimate path lengths for each compartment with the path up to the first branch point labeled as perisomatic (axon: 100 mm, dendrite: 520 mm, perisomatic: 40 mm, ambiguous: 10 mm). We estimated that our skeletons were overestimated by 11% due to following the mesh edges and corrected all reported pathlengths accordingly.

## Code availability

All software is open source and available at http://github.com/seung-lab if not otherwise mentioned.

Alembic: Stitching and alignment.
CloudVolume: Reading and writing volumetric data, meshes, and skeletons to and from the cloud.
Chunkflow: Running convolutional nets on large datasets.
DeepEM: Training convolutional nets to detect neuronal boundaries.
CAVE: Proofreading and connectome updates (visit https://github.com/seung-lab/AnnotationPipelineOverview for repository list).
Igneous: Coordinating downsampling, meshing, and data management.
MeshParty: Interaction with meshes and mesh-based skeletonization (https://github.com/sdorkenw/MeshParty, *Dorkenwald et al., 2020*).
MMAAPP: Watershed, size-dependent single linkage clustering, and mean affinity agglomeration.
PyTorchUtils: Training convolutional nets for synapse detection and partner assignment (https://github.com/nicholasturner1/PyTorchUtils, *Turner, 2021*).
Synaptor: Processing output of the convolutional net for predicting synaptic clefts (https://github.com/nicholasturner1/Synaptor, *Turner et al., 2021*).
TinyBrain and zmesh: Downsampling and meshing (precursors of the libraries that were used).

## Acknowledgements

Supported by the Intelligence Advanced Research Projects Activity (IARPA) via Department of Interior/Interior Business Center (DoI/IBC) contract numbers D16PC00003, D16PC00004, and D16PC0005. The US Government is authorized to reproduce and distribute reprints for Governmental purposes notwithstanding any copyright annotation thereon. HSS also acknowledges support from NIH/NINDS U19 NS104648, ARO W911NF-12-1-0594, NIH/NEI R01 EY027036, NIH/NIMH U01 MH114824, NIH/NINDS R01NS104926, NIH/NIMH RF1MH117815, and the Mathers Foundation, as well as assistance from Google, Amazon, and Intel. We thank S Koolman, M Moore, S Morejohn, B Silverman, K Willie, and R Willie for their image analyses, Garrett McGrath for computer system administration, and May Husseini and Larry and Janet Jackel for project administration. We are grateful to J Maitin-Shepard for neuroglancer and PH Li and V Jain for helpful discussions. We thank DW Tank, K Li, Y Loewenstein, J Kornfeld, A Wanner, M Tsodyks, D Markowitz, and G Ocker for advice and feedback. We thank the Allen Institute for Brain Science founder, Paul G Allen, for his vision, encouragement, and support. Disclaimer: The views and conclusions contained herein are those of the authors and should not be interpreted as necessarily representing the official policies or endorsements, either expressed or implied, of IARPA, DoI/IBC, or the US Government.

# Additional information

## Competing interests

Thomas Macrina, H Sebastian Seung: discloses financial interests in Zetta AI LLC. Jacob Reimer, Andreas S Tolias: discloses financial interests in Vathes LLC. The other authors declare that no competing interests exist.

## Funding

| Funder | Grant reference number | Author |
|---|---|---|
| Intelligence Advanced Research Projects Activity | D16PC00003 | Sven Dorkenwald |
| Intelligence Advanced Research Projects Activity | D16PC00004 | Sven Dorkenwald |
| Intelligence Advanced Research Projects Activity | D16PC00005 | Sven Dorkenwald |
| National Institute of Neurological Disorders and Stroke | U19 NS104648 | H Sebastian Seung |
| Army Research Office | W911NF-12-1-0594 | H Sebastian Seung |
| National Eye Institute | R01 EY027036 | H Sebastian Seung |
| National Institute of Mental Health | U01 MH114824 | H Sebastian Seung |
| National Institute of Neurological Disorders and Stroke | R01 NS104926 | H Sebastian Seung |
| National Institute of Mental Health | RF1MH117815 | H Sebastian Seung |
| G. Harold and Leila Y. Mathers Foundation | | H Sebastian Seung |

The funders had no role in study design, data collection and interpretation, or the decision to submit the work for publication.

## Author contributions

Sven Dorkenwald, Conceptualization, Data curation, Software, Formal analysis, Supervision, Validation, Investigation, Visualization, Methodology, Writing – original draft, Writing – review and editing; Nicholas L Turner, Conceptualization, Data curation, Software, Formal analysis, Validation, Investigation, Visualization, Methodology, Writing – review and editing; Thomas Macrina, Data curation, Software, Supervision, Investigation, Methodology, Project administration, Writing – review and editing; Kisuk Lee, Data curation, Software, Investigation, Visualization, Methodology, Writing – review and editing; Ran Lu, Jingpeng Wu, William M Silversmith, Dodam Ih, Sergiy Popovych, Russel Torres, Gayathri Mahalingam, Yang Li, Data curation, Software; Agnes L Bodor, Adam A Bleckert, Derrick Brittain, Emmanouil Froudarakis, JoAnn Buchanan, Daniel J Bumbarger, Data curation; Nico Kemnitz, Data curation, Software, Visualization, Methodology, Writing – review and editing; Jonathan Zung, Aleksandar Zlateski, Ignacio Tartavull, William Wong, Manuel Castro, Chris S Jordan, Software; Szi-Chieh Yu, Data curation, Supervision; Alyssa M Wilson, Supervision; Marc M Takeno, Data curation, Writing – review and editing; Forrest Collman, Casey M Schneider-Mizell, Data curation, Software, Investigation, Writing – review and editing; Lynne Becker, Shelby Suckow, Project administration; Jacob Reimer, Andreas S Tolias, Supervision, Funding acquisition, Project administration; Nuno Macarico da Costa, Data curation, Supervision, Funding acquisition, Investigation, Project administration, Writing – review and editing; R Clay Reid, Supervision, Funding acquisition, Investigation, Project administration; H Sebastian Seung, Conceptualization, Formal analysis, Supervision, Funding acquisition, Investigation, Methodology, Writing – original draft, Project administration, Writing – review and editing

## Author ORCIDs
Sven Dorkenwald  http://orcid.org/0000-0003-2352-319X
Emmanouil Froudarakis  http://orcid.org/0000-0002-3249-3845
Marc M Takeno  http://orcid.org/0000-0002-8384-7500
Russel Torres  http://orcid.org/0000-0002-2876-4382
Forrest Collman  http://orcid.org/0000-0002-0280-7022
Casey M Schneider-Mizell  http://orcid.org/0000-0001-9477-3853
Nuno Macarico da Costa  http://orcid.org/0000-0003-2001-4568
R Clay Reid  http://orcid.org/0000-0002-8697-6797

## Ethics
All animal procedures were approved by the Institutional Animal Care and Use Committee at the Allen Institute for Brain Science (1503 and 1804) or Baylor College of Medicine (AN-4703).

## Decision letter and Author response
Decision letter https://doi.org/10.7554/eLife.76120.sa1
Author response https://doi.org/10.7554/eLife.76120.sa2

# Additional files

## Supplementary files
• Transparent reporting form

## Data availability
All data acquired and produced for this project are available on https://www.microns-explorer.org/phase1.

The following dataset was generated:

| Author(s) | Year | Dataset title | Dataset URL | Database and Identifier |
| --- | --- | --- | --- | --- |
| Becker L, Bleckert AL, Brittain D, Buchanan J, Bumbarger DJ, Castro M, Cobos E, Collman F, Elabbady L, Dorkenwald S, Froudarakis E, Ih D, Kemnitz N, Jordan CS, Lee K, Li Y, Lu R, MaçaricodaCosta N, Macrina T, Mahalingam G, Mu S, Paninski L, Polleux F, Popovych S, Reid RC, Reimer J, Seung SH, Schneider-Mizell C, Silversmith W, Suckow S, Takeno M, Turner NL, Tartavull I, Tolias AS, Torres R, Wilson AM, Wong W, Wu J, Yang R, S-C Yu, Zhou P, Zlateski A, Zung J | 2020 | MICrONS Layer 2/3 Data Tables | https://doi.org/10.5281/zenodo.5579388 | Zenodo, 10.5281/zenodo.5579388 |

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
