## [Editor Report]

Cortical synaptic plasticity mechanisms shape excitatory connectivity during learning and development. A long-standing question is whether these processes are determined by pre- and postsynaptic activity and whether the resulting synaptic changes result in a continuous, graded distribution of strengths. Dorkenwald and colleagues use extensive ultrastructural data to study cortical excitatory synaptic spines and demonstrate that the population is a very well-described discrete mix of "small" and "large" connections, with graded variability around these dominant modes. Co-innervated connections result in strong correlations between the discrete small/large variable, but not the graded component, supporting a model in which correlated activity results in jumps between small and large synaptic strengths.

---

## [Decision Letter]

**Decision letter after peer review:**

Thank you for submitting your article "Binary and analog variation of synapses between cortical pyramidal neurons" for consideration by *eLife*. Your article has been reviewed by 3 peer reviewers, one of whom is a member of our Board of Reviewing Editors, and the evaluation has been overseen by John Huguenard as the Senior Editor. The following individuals involved in review of your submission have agreed to reveal their identity: Haruo Kasai (Reviewer #2); Thomas M Bartol (Reviewer #3).

Essential revisions:

1) Mechanistic claims, statistical analyses and their interpretation: please take on board reviewer's comments about the causal interpretations of these findings and adjust the claims and language in the manuscript to acknowledge that the findings are observational. In several places the writing appears to claim that statistical analyses alone can reveal causal mechanisms, which is misleading. Please revise the writing to more clearly delineate hypotheses and interpretations from empirical observations. Finally, address Reviewer #1's concern about bias in the analysis of size distribution.

2) Potential control/comparative data: consider suggestions from Reviewer #3 for comparative analysis of other synapse types that may be available within the dataset. If these data are not readily available please provide a brief explanation in a rebuttal.

3) Literature: review the paper's treatment of previous work in line with some of the suggestions made by reviewer #2. You do not have to adopt the reviewer's interpretations of the data, but it is worth reviewing some of the suggested references to ensure the manuscript does justice to existing work and current thinking on the relationship between plasticity mechanisms and synapse size distributions.

*Reviewer #1 (Recommendations for the authors):*

I think the paper is excellent overall and I only have two comments about the wording and interpretation of some of the findings and about one part of the statistical analysis.

1. Interpretations. In several places in the manuscript the authors make statements about the findings being biologically meaningful *as opposed to being statistical observations*. The fact is there are compelling observations and sound analyses and interpretations of these observations, but this does not transform the study from being observational in nature. A key example is in the paragraph at the top of page 8:

"A binary mixture model might merely be a convenient way of approximating deviations from normality. We would like to know whether the components of our binary mixture really have a biological basis, i.e., whether they correspond to two structural states of synapses. A mixture of two normal distributions can be unimodal or bimodal, depending on the model parameters3 (Robertson and Fryer, 1969). When comparing best fit unimodal and bimodal mixtures we found that a bimodal model yields a significantly superior fit for spine volume and geometric mean of spine volume (p=0.0425, n=320; Extended Data Figure 4, see (Holzmann and Vollmer, 2008) for statistical methods). This bimodality makes it plausible that the mixture components correspond to biological states of synapses."

The analyses they have performed to test for bimodality and the interpretation they have for its presence are both sound. However, the authors are saying that statistical modelling alone can reveal whether measurements "really have a biological basis". It cannot. Only a further (experimental) intervention could begin to do this. I want to say very clearly that I don't think the observational nature of the study is a weak point at all. The authors should accept it is observational and describe it as such. They have produced a very compelling and sophisticated piece of science, but it is incorrect (and slightly dangerous for more naïve readers) to blur the lines between the reasoning that motivates a hypothesis, and the statistical means of evaluating evidence for it.

I would hazard a guess that the authors are aware of this issue but are dealing with the rather unscientific way our community treats/labels observational work. I think this is excellent work and it doesn't need to disguise the epistemology at all.

Finally, I'd like to point out that a bimodal distribution is still a 'continuum'. The authors don't contradict this, but they come close to contrasting continual synaptic variation with their findings in the abstract which I find potentially misleading if readers are sloppy.

2. Statistical analysis. I have a simple query about the way the authors rule out a trend between the analog component of spine volume correlation between multiple connections (Figure 4 and associated analyses).

My understanding is that they resampled from the data based on mixture component weight (i.e. preferentially sample from points close to an effective cluster) THEN perform correlation analysis on these two resampled populations.

Is this biased? i.e. would it tend to dilute residual correlation that cannot be accounted for by the binary components because it over-represents data points close to the centroids of clusters? I'm not sure, but I think they could check this through simulation very easily.

An alternative method for asking about 'analog covariation' would be to simply look at residuals of the model with the binary component subtracted, as is done in standard mixed statistical models. In this case a significant trend in the residuals would be evidence for analog covariation.

Finally, the pedant in me wants them to be more careful about absence of evidence not being evidence for absence, so they could tighten their language in places when describing these results in the event of a robust null finding.

*Reviewer #2 (Recommendations for the authors):*

1) Abstract: "Previous cortical studies modelled a continuum of synapse sizes (Arellano et al., 2007)”

The continuum of spine sizes has already been shown in old ssEM papers, such as Harris, K. M. and Stevens, J Neurosci 9, 2982-2997 (1989). The same applied to all the rest of the text.

2) Abstract: "by a log-normal distribution (Loewenstein, Kuras and Rumpel, 2011; de Vivo et al., 2017; Santuy et al. , 2018)".

These papers do not provide a rationale for a log-normal distribution of spine sizes and are misleading. There is no reason and evidence why spine distribution is log-normal. Note that the multiplicative dynamics do not simply predict a log-normal distribution. The most comprehensive review on an approximately log-normal distribution has been provided in the following review, which should be cited to help readers: Kasai, H., Ziv, N. E., Okazaki, H., Yagishita, S. and Toyoizumi, Nature reviews. Neuroscience 22, 407-422, doi:10.1038/s41583-021-00467-3 (2021).

3) Introduction: "In the 2000s, some hypothesized that long-term plasticity involves discrete transitions of synapses between two structural states (Kasai et al., 2003; Bourne and Harris, 2007)."

This sentence is wrong. None of the two papers claimed the discrete transition of synapses between two states. They describe the spines as a continuum, emphasizing that learning changes smaller spines to bigger ones, consistent with the binary and analogue variation of synapses proposed in this study.

Bistability was predicted only theoretically as winner-takes-all situations, ex. Gilson, M. and Fukai, T. PloS one 6, e25339, doi:10.1371/journal.pone.0025339 (2011).

4) Page 7 "Even researchers who report bimodally distributed synapse size in the hippocampus (Spano et al., 2019) still find log-normally distributed synapse size in the neocortex (de Vivo et al., 2017) by the same methods."

These statements make sense only when the spine volumes are plotted on a logarithmic scale. The authors should consider using "bimodal on the semi-logarithmic scale" whenever the bimodality matters. Also, by comparing Figure 3b and c, the authors should explicitly describe that the bimodality only becomes evident when a semi-logarithmic plot is used.

5) The authors should display the linear plots also for Figure 1d and 1e.

6) The authors should provide more detailed descriptions of the behavioural states of mice, as the results should depend on how mice were rared. Say, there should be more binary mode on a semi-logarithmic plot in mice rared in an environment enriched cage.

7) Methods section states two-photon imaging, but the study does not seem to use two-photon data.

8) Discussion: "Experiments have shown that large dynamical fluctuations persist even after activity is pharmacologically blocked (Yasumatsu et al. , 2008; Statman et al. , 2014)."

They are also supported by more recent data by Sigler et al. Neuron 94:304(2017) and Sando et al. Neuron 94:312(2017).

9) Discussion: "It has been argued that the observed structural volatility of synapses is challenging to reconcile with the stability of memory (Loewenstein, Kuras and Rumpel, 2011). Our findings suggest two possible resolutions of the stability-plasticity dilemma……. In a second scenario…"

These discussions do not provide a resolution. As described in Kasai et al. (Nat Rev Neurosci 2021), we should be aware that most daily memories are forgotten in a few days to 1 week, and longer-lasting memories need repeated recall, as initially described by Ebbinghaus (1885). There is no stability-plasticity dilemma when we take these memory properties into account. The spine fluctuations also naturally explain the memory persistence and spine volume distributions. The author should rewrite the discussion incorporating this coherent view.

9) The authors find that the dual connections by the axons from outside the 250*140*90um volume were not bimodal in the semi-logarithmic plot (EFigure 10), suggesting that a cell assembly is more often formed within the volume than distant cortices. The authors should explicitly describe and discuss this scenario.

*Reviewer #3 (Recommendations for the authors):*

1) In the Abstract it would helpful to state the animal and cortical region studied and the size of the dataset (volume, number of connections).

2) Near the end of Abstract, perhaps give a few examples of the "other influences" that contribute to the analog variation of synapse size in dual connections.

3) At the end of the Abstract, "stability-plasticity dilemma" might be a bit vague for some readers.

4) In Introduction, first paragraph, "Spine dynamics were interpreted as synaptic plasticity" is an odd statement since "dynamics" means change and "plasticity" means change. Please reword.

5) In Introduction, paragraph 4, the authors seem to equate their definition of "paired connections" (or "dual connections") with that used in Bartol et al., 2015. The authors should clearly define their use of the term "paired (or dual) connection" and the definition of "Same Dendrite, Same Axon pairs (SDSA pairs)" used in Bartol et al., 2015. This distinction is important and could explain some of the differences observed in their new results here compared to the earlier observations in the literature.

6) Bartol et al. 2015 showed that the sizes of SDSA pairs in hippocampus are highly correlated along the whole continuum with no binary component. Which differs from the authors' results. This difference is interesting for further discussion.

7) Introduction paragraph 5, please be specific about the "specificity of the synaptic population". Please state again that these are connections between L2/3 PyCs. Also, again Bartol et al. 2015 showed strong corellation in the continuum among all SDSA pairs, from smallest to largest, though in Rat hippocampus, not Mouse neocortex.

8) Handling of Image Defects. In discussing the defects it helps the reader to give the xyz resolution of the ssEM images here. 3.58 nm in plane, 40 nm axial.

9) Page 7, paragraph 7, please be more clear about parallel and serial multisynaptic connections. By parallel do you mean multiple synapses made by separate branches of branching axons? And would series mean en-passant synapses of a single stretch of axon? Please note that the SDSA pairs of Bartol et al., were always en-passant synapses of single axons on to the same dendritic branch within just a few microns, not different branches.

10) Page 9 last paragraph, please be clear about dual connections vs. SDSA pairs here.

11) Page 10 last paragraph, by separation distance do you mean along the same dendrite, different dendrites, same axon, different axons, Euclidean distance in the volume?

12) Page 11, second paragraph, why not draw random pairs from the whole set, n=1960, of synapses?

13) Discussion, paragraph 1, again please don't equate your dual connections with the SDSA pairs of Bartol et al., 2015.

14) Discussion, paragraph 3, there could also be differences in synaptic plasticity mechanisms in different brain regions and cell types, neural subcircuits, etc…

---

## [Author Response]

Essential revisions:1) Mechanistic claims, statistical analyses and their interpretation: please take on board reviewer's comments about the causal interpretations of these findings and adjust the claims and language in the manuscript to acknowledge that the findings are observational. In several places the writing appears to claim that statistical analyses alone can reveal causal mechanisms, which is misleading. Please revise the writing to more clearly delineate hypotheses and interpretations from empirical observations. Finally, address Reviewer #1's concern about bias in the analysis of size distribution.

We adjusted the language accordingly and addressed reviewer #1’s concern about bias in the analysis.

2) Potential control/comparative data: consider suggestions from Reviewer #3 for comparative analysis of other synapse types that may be available within the dataset. If these data are not readily available please provide a brief explanation in a rebuttal.

We included a comparison with synapses with inhibitory neurons in the dataset and extended Figure 2.

3) Literature: review the paper's treatment of previous work in line with some of the suggestions made by reviewer #2. You do not have to adopt the reviewer's interpretations of the data, but it is worth reviewing some of the suggested references to ensure the manuscript does justice to existing work and current thinking on the relationship between plasticity mechanisms and synapse size distributions.

We included new citations as suggested by the reviewers and adjusted the text to more reflect the reviewer’s interpretations of the data.

Reviewer #1 (Recommendations for the authors):I think the paper is excellent overall and I only have two comments about the wording and interpretation of some of the findings and about one part of the statistical analysis.1. Interpretations. In several places in the manuscript the authors make statements about the findings being biologically meaningful *as opposed to being statistical observations*. The fact is there are compelling observations and sound analyses and interpretations of these observations, but this does not transform the study from being observational in nature. A key example is in the paragraph at the top of page 8:"A binary mixture model might merely be a convenient way of approximating deviations from normality. We would like to know whether the components of our binary mixture really have a biological basis, i.e., whether they correspond to two structural states of synapses. A mixture of two normal distributions can be unimodal or bimodal, depending on the model parameters3 (Robertson and Fryer, 1969). When comparing best fit unimodal and bimodal mixtures we found that a bimodal model yields a significantly superior fit for spine volume and geometric mean of spine volume (p=0.0425, n=320; Extended Data Figure 4, see (Holzmann and Vollmer, 2008) for statistical methods). This bimodality makes it plausible that the mixture components correspond to biological states of synapses."The analyses they have performed to test for bimodality and the interpretation they have for its presence are both sound. However, the authors are saying that statistical modelling alone can reveal whether measurements "really have a biological basis". It cannot. Only a further (experimental) intervention could begin to do this. I want to say very clearly that I don't think the observational nature of the study is a weak point at all. The authors should accept it is observational and describe it as such. They have produced a very compelling and sophisticated piece of science, but it is incorrect (and slightly dangerous for more naïve readers) to blur the lines between the reasoning that motivates a hypothesis, and the statistical means of evaluating evidence for it.

We have toned down the biological interpretation.

I would hazard a guess that the authors are aware of this issue but are dealing with the rather unscientific way our community treats/labels observational work. I think this is excellent work and it doesn't need to disguise the epistemology at all.Finally, I'd like to point out that a bimodal distribution is still a 'continuum'. The authors don't contradict this, but they come close to contrasting continual synaptic variation with their findings in the abstract which I find potentially misleading if readers are sloppy.2. Statistical analysis. I have a simple query about the way the authors rule out a trend between the analog component of spine volume correlation between multiple connections (Figure 4 and associated analyses).My understanding is that they resampled from the data based on mixture component weight (i.e. preferentially sample from points close to an effective cluster) THEN perform correlation analysis on these two resampled populations.Is this biased? i.e. would it tend to dilute residual correlation that cannot be accounted for by the binary components because it over-represents data points close to the centroids of clusters? I'm not sure, but I think they could check this through simulation very easily.An alternative method for asking about 'analog covariation' would be to simply look at residuals of the model with the binary component subtracted, as is done in standard mixed statistical models. In this case a significant trend in the residuals would be evidence for analog covariation.

We re-analyzed the set of 160 synaptic pairs from dual-synaptic connections accordingly. We assigned pairs to their most likely state (SS, SL, LS, LL), subtracted the mean of the assigned state and plotted the residuals in Figure 4—figure supplement 3. We repeated this analysis while restricting assignments to SS and LL. We did not find a significant correlation in the residuals.

Finally, the pedant in me wants them to be more careful about absence of evidence not being evidence for absence, so they could tighten their language in places when describing these results in the event of a robust null finding.Reviewer #2 (Recommendations for the authors):1) Abstract: "Previous cortical studies modelled a continuum of synapse sizes (Arellano et al., 2007)The continuum of spine sizes has already been shown in old ssEM papers, such as Harris, K. M. and Stevens, J Neurosci 9, 2982-2997 (1989). The same applied to all the rest of the text.

We now cite this paper throughout the manuscript.

2) Abstract: "by a log-normal distribution (Loewenstein, Kuras and Rumpel, 2011; de Vivo et al., 2017; Santuy et al. , 2018)".These papers do not provide a rationale for a log-normal distribution of spine sizes and are misleading. There is no reason and evidence why spine distribution is log-normal. Note that the multiplicative dynamics do not simply predict a log-normal distribution. The most comprehensive review on an approximately log-normal distribution has been provided in the following review, which should be cited to help readers: Kasai, H., Ziv, N. E., Okazaki, H., Yagishita, S. and Toyoizumi, Nature reviews. Neuroscience 22, 407-422, doi:10.1038/s41583-021-00467-3 (2021).

We appreciate the suggested reference and added it to the manuscript. The original language “well-modeled by a log-normal distribution” has been toned down to “approximated by a log-normal distribution.”

Our original text does not mention multiplicative dynamics. We have added a footnote saying that there are dynamical models that yield approximately log-normal distributions, with a reference to the 2021 review.

3) Introduction: "In the 2000s, some hypothesized that long-term plasticity involves discrete transitions of synapses between two structural states (Kasai et al., 2003; Bourne and Harris, 2007)."This sentence is wrong. None of the two papers claimed the discrete transition of synapses between two states. They describe the spines as a continuum, emphasizing that learning changes smaller spines to bigger ones, consistent with the binary and analogue variation of synapses proposed in this study.Bistability was predicted only theoretically as winner-takes-all situations, ex. Gilson, M. and Fukai, T. PloS one 6, e25339, doi:10.1371/journal.pone.0025339 (2011).

We have changed the text to read, “In the 2000s, some hypothesized the existence of “learning spines” and “memory spines,” appearing to define two discrete categories that are structurally and functionally different.” We hope that this description of the two papers is accurate.

4) Page 7 "Even researchers who report bimodally distributed synapse size in the hippocampus (Spano et al., 2019) still find log-normally distributed synapse size in the neocortex (de Vivo et al., 2017) by the same methods."These statements make sense only when the spine volumes are plotted on a logarithmic scale. The authors should consider using "bimodal on the semi-logarithmic scale" whenever the bimodality matters. Also, by comparing Figure 3b and c, the authors should explicitly describe that the bimodality only becomes evident when a semi-logarithmic plot is used.

We adjusted the text in several places to now clarify that the observation of the bimodality requires the log-scale.

5) The authors should display the linear plots also for Figure 1d and 1e.

We added the linear plots for Figures 3d and e (Figure 3—figure supplement 1).

6) The authors should provide more detailed descriptions of the behavioural states of mice, as the results should depend on how mice were rared. Say, there should be more binary mode on a semi-logarithmic plot in mice rared in an environment enriched cage.

We added a description of the upbringing of the mouse to the methods section.

7) Methods section states two-photon imaging, but the study does not seem to use two-photon data.

We added the two-photon imaging to the methods section to give context about the experimental circumstances of the mice prior to EM acquisition to the reader. The two-photon data was not used in this paper.

8) Discussion: "Experiments have shown that large dynamical fluctuations persist even after activity is pharmacologically blocked (Yasumatsu et al. , 2008; Statman et al. , 2014)."They are also supported by more recent data by Sigler et al. Neuron 94:304(2017) and Sando et al. Neuron 94:312(2017).

We appreciate the suggested references and added them to the manuscript.

9) Discussion: "It has been argued that the observed structural volatility of synapses is challenging to reconcile with the stability of memory (Loewenstein, Kuras and Rumpel, 2011). Our findings suggest two possible resolutions of the stability-plasticity dilemma……. In a second scenario…"These discussions do not provide a resolution. As described in Kasai et al. (Nat Rev Neurosci 2021), we should be aware that most daily memories are forgotten in a few days to 1 week, and longer-lasting memories need repeated recall, as initially described by Ebbinghaus (1885). There is no stability-plasticity dilemma when we take these memory properties into account. The spine fluctuations also naturally explain the memory persistence and spine volume distributions. The author should rewrite the discussion incorporating this coherent view.

We have removed the sentence about “resolutions of the stability-plasticity dilemma,” which in retrospect was perhaps too sweeping a claim.

10) The authors find that the dual connections by the axons from outside the 250*140*90um volume were not bimodal in the semi-logarithmic plot (EFigure 10), suggesting that a cell assembly is more often formed within the volume than distant cortices. The authors should explicitly describe and discuss this scenario.

Another possible explanation is that observation of bimodality requires restricting to synapses between a particular cell type, e.g., L2/3 pyramidal neurons in this case. We have added a new paragraph to clarify this idea.

Reviewer #3 (Recommendations for the authors):1) In the Abstract it would helpful to state the animal and cortical region studied and the size of the dataset (volume, number of connections).

We added volume, region, animal and number information to the abstract.

2) Near the end of Abstract, perhaps give a few examples of the "other influences" that contribute to the analog variation of synapse size in dual connections.3) At the end of the Abstract, "stability-plasticity dilemma" might be a bit vague for some readers.

We replaced this sentence with “implications for the longstanding hypothesis that activity-dependent plasticity switches synapses between bistable states.”

4) In Introduction, first paragraph, "Spine dynamics were interpreted as synaptic plasticity" is an odd statement since "dynamics" means change and "plasticity" means change. Please reword.

We see the reviewers' concern about redundant phrasing. In this sentence we linked the terms “dynamic” and “plasticity” to different structures.

5) In Introduction, paragraph 4, the authors seem to equate their definition of "paired connections" (or "dual connections") with that used in Bartol et al., 2015. The authors should clearly define their use of the term "paired (or dual) connection" and the definition of "Same Dendrite, Same Axon pairs (SDSA pairs)" used in Bartol et al., 2015. This distinction is important and could explain some of the differences observed in their new results here compared to the earlier observations in the literature.6) Bartol et al. 2015 showed that the sizes of SDSA pairs in hippocampus are highly correlated along the whole continuum with no binary component. Which differs from the authors' results. This difference is interesting for further discussion.7) Introduction paragraph 5, please be specific about the "specificity of the synaptic population". Please state again that these are connections between L2/3 PyCs. Also, again Bartol et al. 2015 showed strong corellation in the continuum among all SDSA pairs, from smallest to largest, though in Rat hippocampus, not Mouse neocortex.

We thank the reviewer for highlighting the problem of ambiguity when comparing the work of Bartol et al.. We now differentiate our work from previous work more clearly by clarifying the difference in the studied synapse populations.

8) Handling of Image Defects. In discussing the defects it helps the reader to give the xyz resolution of the ssEM images here. 3.58 nm in plane, 40 nm axial.

We added the resolution to the text as suggested.

9) Page 7, paragraph 7, please be more clear about parallel and serial multisynaptic connections. By parallel do you mean multiple synapses made by separate branches of branching axons? And would series mean en-passant synapses of a single stretch of axon? Please note that the SDSA pairs of Bartol et al., were always en-passant synapses of single axons on to the same dendritic branch within just a few microns, not different branches.

We clarified this in the text.

10) Page 9 last paragraph, please be clear about dual connections vs. SDSA pairs here.

We clarified this in the text.

11) Page 10 last paragraph, by separation distance do you mean along the same dendrite, different dendrites, same axon, different axons, Euclidean distance in the volume?

We clarified this in the text. We referred to the median euclidean distance in the volume.

12) Page 11, second paragraph, why not draw random pairs from the whole set, n=1960, of synapses?

Unfortunately, we could not find out what part of the analysis this comment is referring to.

13) Discussion, paragraph 1, again please don't equate your dual connections with the SDSA pairs of Bartol et al., 2015.

We incorporated this comment by weakening the comparative language.

14) Discussion, paragraph 3, there could also be differences in synaptic plasticity mechanisms in different brain regions and cell types, neural subcircuits, etc…

We thank the reviewers for their feedback and suggestions!